# Exploiting weight-space symmetries for approximating curvature

Artem Artemev [1]   Rui Xia [2 1]   Benjamin M. Boyd [2 1]   Youjing Yu [2 1]   Felix Dangel [3]   Guillaume Hennequin [2 1]
Alberto Bernacchia [1]

## Abstract

Many machine learning techniques rely on approximating a loss function's curvature, but this is notoriously hard to do at the scale of modern deep networks. Surprisingly, no previous work has exploited the curvature constraints that arise from well known weight-space symmetries in loss landscapes. By analytically averaging over group actions that leave the loss invariant, we construct structured Hessian approximations from single gradients that can be tractably estimated, stored, and inverted. The choice of user-specified symmetry group directly governs the trade-off between approximation accuracy and computational cost. Moreover, our framework provides a unifying theoretical lens for viewing existing methods; in particular, a specific choice of symmetry group recovers Shampoo/Muon-like curvature estimates. We validate our method on a range of network architectures, and deploy it to second-order optimization benchmarks, including a small language model. Our curvature estimation framework might find applications in other machine learning problems such as uncertainty estimation, continual learning, compression/pruning, training data attribution, and more.

## 1. Introduction

Efficient estimation of a loss function's (inverse) curvature has become a pillar of many machine learning sub-fields. In second-order optimization, an accurate and tractable estimate of curvature may be used to precondition gradient-based updates and obtain faster convergence (Bottou et al., 2018). In Bayesian deep learning, it is the core ingredient in the Laplace approximation of the posterior over weights (Mackay, 1992; Ritter et al., 2018a). In continual learning, it

[1]MediaTek, Cambridge, UK [2]University of Cambridge, Cambridge, UK [3]Vector Institute, Toronto, Canada. Correspondence to: Artem Artemev <art.art.v@gmail.com>.

*Proceedings of the 43rd International Conference on Machine Learning*, Seoul, South Korea. PMLR 306, 2026. Copyright 2026 by the author(s).

can help identify directions in parameter space along which previously acquired knowledge is protected (Ritter et al., 2018b; Kao et al., 2021). Likewise, some network compression algorithms rely on second-order information to score the importance of weights and prioritize pruning (McGowan et al., 2024; Wang et al., 2019; Theis et al., 2018; Kurtic et al., 2022; Hassibi and Stork, 1992).

Despite the ubiquitous usefulness of second-order information, the computational and memory requirements of constructing, storing, and inverting curvature matrices are often prohibitive. The field of second-order optimization has been a useful incubator for many commonly used approximation techniques that scale to large models. Optimizers such as (E)KFAC (Martens and Grosse, 2015; George et al., 2018), Shampoo (Gupta et al., 2018) or Soap (Vyas et al., 2025) approximate curvature in positive-definite, block-diagonal form where each block is Kronecker-factorized. This leads to efficient estimation, storage, and inversion. These methods have proven remarkably effective in large-scale applications (Kasimbeg et al., 2025; Liu et al., 2025; Grosse et al., 2023) and have been mathematically dissected from multiple angles (Buffelli et al., 2024; Frans et al., 2025; Lin et al., 2025; Eschenhagen et al., 2025; Pethick et al., 2025a).

Here, we present a different approach for approximating curvature, rooted in a key insight recently highlighted in a few studies (Bernacchia, 2025; Ainsworth et al., 2022; Rossi et al., 2023; Entezari et al., 2022): many loss functions in deep learning are invariant under certain transformations of their parameters, such as rescaling, permutations, or broader orthogonal transformations (Hecht-Nielsen, 1990; Chen et al., 1993; see Fig. 1, top, for a permutation example). Identifying such groups is often straightforward. Critically, loss invariance implies gradient equivariance, such that knowing a single gradient is enough to analytically compute the gradients at many other points across the loss landscape, namely those reachable by a symmetry transformation (the *group orbit*, Fig. 1, bottom). Our first contribution is to show how to analytically distill these gradient variations across the orbit into an estimate of the loss curvature – using a single gradient computation. The solution involves the orbit averages of the gradient which belongs to the commutant algebra (Definition D.4) of the given symmetry group, and is therefore provably highly structured: a

linear superposition of a small number of basis elements. This implies that our approximate Hessian can be represented and stored in the compact form of the "factors" that weigh the contribution of each basis element in the algebra. We provide a just-in-time compiler that turns user-provided, symbolic specifications of symmetries into efficient PyTorch routines computing the corresponding factors.

Our work sheds new light on the accuracy-complexity trade-off inherent in the art of approximating Hessians. For a large symmetry group, the commutant algebra is very low-dimensional, resulting in cheap calculations. However, the associated group orbits stretch very far in parameter space, with gradient variations along the orbit eventually ceasing to reflect local curvature. Conversely, small symmetry groups have smaller, more "local" orbits, but yield more expensive Hessian approximations that comprise more terms. We show that both Shampoo and Muon can be understood as making a specific choice of permutation subgroup within our framework; this choice appears to hit a sweet spot in complexity vs. performance. More generally, our framework provides a systematic recipe for constructing symmetry-aware Hessian approximations that fit varying computational budgets.

In summary, we contribute the following:

- We provide a novel Hessian approximation, derived from a single gradient computation by averaging over the gradient's symmetry group orbit (Sec. 3.1).

- Using multi-layer perceptrons (MLPs) and Transformers as illustrative examples, we show theoretically that our Hessian approximation's complexity depends on the symmetry group size, and that the group can be restricted to match a desired complexity (Sec. 3.2).

- We apply our approximate Hessian to second-order optimization, and we prove mathematically that a specific choice of the symmetry group corresponds to well-known optimizers, Shampoo and Muon (Sec. 3.4).

- We test empirically the quality of the Hessian approximation and the equivalence with Shampoo/Muon, on MLP and Transformer architectures (Secs. 4 and 5).

## 1.1. Related Work

The ubiquity of weight-space symmetries in neural networks is a well-known fact (Ravanbakhsh et al., 2017; Ziyin et al., 2024; Zhao et al., 2023). Symmetries have been exploited to improve optimization (Neyshabur et al., 2015; Meng et al., 2018; Zhao et al., 2022), to analyze the structure and stability of fixed points in the loss landscape (Fukumizu and Amari, 2000; Simsek et al., 2021; Zhang et al., 2021), and to justify model merging (Entezari et al., 2022; Ainsworth et al., 2022).

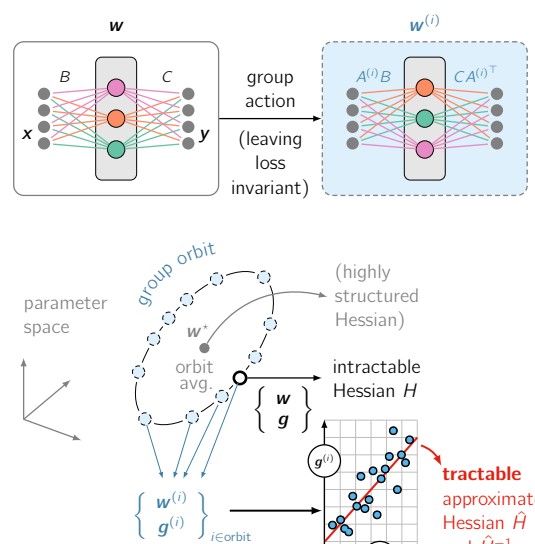

*Figure 1.* We present a framework for tractably approximating Hessians of large models by harnessing their inherent symmetries. **Top**: neural networks are typically invariant to many types of transformations, such as permutation of units within each layer (Sec. 2.1). **Bottom:** given a current parameter set $w$ and associated loss gradient $g$, all similar pairs $(w^{(i)}, g^{(i)})$ obtained by symmetry form the *group orbit* and are analytically accessible, as loss invariance implies gradient equivariance (Sec. 2.2). This large set describes how gradients vary across the parameter space. Here, we show that such curvature information can be implicitly manipulated and summarized into a tractable approximation $\hat{H}$ to the intractable Hessian $H$ (Sec. 3.1).

Surprisingly, there has been little work on how to exploit such symmetries for constraining curvature estimates. Kunin et al. (2020) and Ziyin (2023) showed that symmetries place structural constraints on the Hessian, with the latter proving that at symmetric critical points the Hessian inherits the symmetry of the group. Our approximations share this structure at any point during training, and can be used to estimate curvature from a single gradient.

Most related to ours is the work of Bernacchia (2025), who showed how to construct a form of "global curvature" by analytically averaging gradients across an ensemble of neural networks, which depends on the weights initialization. Here, we arrive at a similar mathematical construction but from the principled angle of Hessian approximation in a *single model*, irrespective of how the model is initialized, and we extend the theory beyond MLPs to Transformer models.

A few previous studies provided theoretical interpretations of Shampoo and Muon. Recent work showed that such optimizers can be interpreted as an approximate version of second-order optimization, the Gauss-Newton method (Morwani et al., 2025). Another line of work argues that they are equivalent to the spectral descent method (Bernstein and Newhouse, 2024; Pethick et al., 2025a). Our work

establishes a new connection between those optimizers and the model architecture's symmetries.

## 2. Background and notation

### 2.1. Invariance to weight-space transformations

Most neural networks parameterize a loss function $\mathcal{L}(\boldsymbol{w})$ that is invariant to specific parameter transformations, or group actions. Consider the toy MLP network shown in Fig. 1 (top) where $B$ and $C$ are the first and second layer weight matrices, respectively, and input-output map $\boldsymbol{y} = C\phi(B\boldsymbol{x})$, where $\phi(\cdot)$ is an element-wise non-linearity. This map is invariant to any simultaneous permutation of the rows of $B$ and the columns of $C$. That is, for any permutation matrix $A_1$, the transformations $B \to A_1 B$ and $C \to C A_1^\top$ leaves $\boldsymbol{y}$ unaffected. We refer to $A_1$ as the matrix representation of an element of the symmetric group of permutations, denoted by $\mathcal{G}_1$.

We denote by $\boldsymbol{w} = [\text{vec}(B); \text{vec}(C)]$ the vectorized tensor of parameters. In terms of vectorized tensors $\boldsymbol{b} = \text{vec}(B)$ and $\boldsymbol{c} = \text{vec}(C)$, those transformations are rewritten, equivalently, as $\boldsymbol{b} \to (\mathrm{I} \otimes A_1)\, \boldsymbol{b}$, and $\boldsymbol{c} \to (A_1 \otimes \mathrm{I})\, \boldsymbol{c}$, with the Kronecker product $\otimes$. The identity matrix $\mathrm{I}$ represents the fact that the input and output components of the neural net cannot be transformed without affecting the map. For convenience of notation, we define $\mathcal{G}_0$ to be the trivial group with the identity element only, $A_0 = \mathrm{I}$, and we rewrite the transformations as $\boldsymbol{b} \to (A_0 \otimes A_1)\, \boldsymbol{b}$ and $\boldsymbol{c} \to (A_1 \otimes A_0)\, \boldsymbol{c}$.

More generally, a network parameterized by a set of tensors might be invariant to transformations applied to specific axes of specific tensors. We denote by $\mathcal{G} = \mathcal{G}_0 \times \mathcal{G}_1 \times \ldots \times \mathcal{G}_m$ the direct product of all groups acting on different tensors, and by $A_i$ the matrix representation of a member of group $\mathcal{G}_i$. For a vectorized tensor $\boldsymbol{v}$ of order $d$ within the network's parameter set, we use the Kronecker notation $\boldsymbol{v} \to \left(\bigotimes_{k=1}^{d} A_{i(k)}\right)\boldsymbol{v}$ to express that the tensor is transformed by $A_{i(k)}$ along its $k^{\text{th}}$ axis. The notation $i(k)$ allows for the possibility that axes of different tensors are tied to the same group action, as in the case of the 2-layer MLP, with $i(1) = 0$, $i(2) = 1$ for $\boldsymbol{b}$, and $i(1) = 1$, $i(2) = 0$ for $\boldsymbol{c}$.

In most applications, invariance of the network's input-output map implies invariance of the loss. An important consequence of that is gradient equivariance: if $\mathcal{L}(\boldsymbol{w}) = \mathcal{L}(A\boldsymbol{w})$ for some orthogonal parameter transformation $A$, then a straightforward application of the chain rule, together with $A^{-\top} = A$, yields $\nabla\mathcal{L}(A\boldsymbol{w}) = A\nabla\mathcal{L}(\boldsymbol{w})$.

### 2.2. Orbit averages

Much of our theory is based on the concept of orbit averaging, i.e. averaging specific expressions over all elements of the relevant group (Fulton and Harris, 2013). For an order-$d$

(vectorized) tensor $\boldsymbol{v}$, we consider the first-order average

$$\mathcal{R}_1(\boldsymbol{v}, \mathcal{G}) \equiv \mathbb{E}_{\mathcal{G}}\left[\left(\bigotimes_{k=1}^{d} A_{i(k)}\right)\boldsymbol{v}\right] \tag{1}$$

where $\mathbb{E}_{\mathcal{G}}[\cdot]$ denotes an average over all elements in $\mathcal{G}$, i.e. an average over all possible choices of $A_1, A_2, \ldots$. Similarly, we will need the (second-order) orbit average of the outer product of two (vectorized) tensors:

$$\mathcal{R}_2(\boldsymbol{v}, \boldsymbol{v}', \mathcal{G}) \equiv$$
$$\mathbb{E}_{\mathcal{G}}\left[\left(\bigotimes_{k=1}^{d} A_{i(k)}\right)\boldsymbol{v}\boldsymbol{v}'^\top\left(\bigotimes_{k=1}^{d'} A_{i'(k)}\right)^\top\right] \tag{2}$$

where $\boldsymbol{v}'$ is another vectorized tensor of order $d'$.

### 2.3. Shampoo and Muon

Shampoo (Gupta et al., 2018) is a second-order optimizer that approximates the loss curvature in a structured way to construct a gradient preconditioner of each parameter tensor in a neural network. Specifically, for a parameter matrix $W_t$ and corresponding loss gradient $G_t$, Shampoo computes two running sums across training iterations $t$:

$$L_{t+1} = L_t + G_t G_t^\top, \qquad (L_0 = \epsilon I) \tag{3a}$$
$$R_{t+1} = R_t + G_t^\top G_t. \qquad (R_0 = \epsilon I) \tag{3b}$$

Some versions of Shampoo implement an exponential moving average instead of temporal accumulation (Eschenhagen et al., 2025; Shi et al., 2023). The weight update is equal to

$$W_{t+1} = W_t - \eta L_t^{-\frac{1}{4}} G_t R_t^{-\frac{1}{4}} \tag{4}$$

where $\eta$ is a learning rate. This update can be viewed as an approximation to the standard second-order update where the Hessian has block-diagonal form, and each block is equal to $R_t^{1/4} \otimes L_t^{1/4}$ (Morwani et al., 2025).

The Shampoo optimizer is closely related to the recently proposed Muon optimizer (Jordan et al., 2024). Leaving aside the temporal accumulation of $L_t$ and $R_t$, by using a singular value decomposition $G_t = U\Sigma V^\top$, one can show

$$L_t^{-\frac{1}{4}} G_t R_t^{-\frac{1}{4}} \sim \left(G_t G_t^\top\right)^{-\frac{1}{4}} G_t \left(G_t^\top G_t\right)^{-\frac{1}{4}} = UV^\top, \tag{5}$$

with further details in Appendix H.1. This uniformization of gradient singular values, or 'polar decomposition', is the essence of Muon. Muon approximates this by running a few iterations of the Newton-Schulz algorithm, which scales better than Shampoo's SVD-based computation of $L_t^{-1/4}$ and $R_t^{-1/4}$. Despite recent progress (Morwani et al., 2025; Bernstein and Newhouse, 2024; Pethick et al., 2025b), we still lack a principled understanding of why Shampoo and Muon optimizers have proven so effective.

# 3. Theoretical results

In this section, we provide our main theoretical contributions. In Sec. 3.1, we derive our novel approximation of the Hessian, based on orbit group averaging of gradients. In Sec. 3.2, we use MLPs and Transformers as examples to show that the complexity of the Hessian depends on the size of the symmetry group. In Sec. 3.4, we prove that Shampoo/Muon optimizers correspond to an intermediate size of the symmetry group.

Before we start, we state the following Lemma, which is instrumental in deriving many of our results and is proven in Appendix A:

**Lemma 3.1.** *For a given group $\mathcal{G}$ and the matrix representation $A$ of any element in the group, the orbit average is invariant (for all $v$, $v'$)*

$$A\,\mathcal{R}_1(v, \mathcal{G}) = \mathcal{R}_1(v, \mathcal{G}), \tag{6a}$$

$$A\,\mathcal{R}_2(v, v', \mathcal{G})A^\top = \mathcal{R}_2(v, v', \mathcal{G}). \tag{6b}$$

## 3.1. Using orbit averages to approximate curvature

The invariance of a neural network to a given group $\mathcal{G}$ of transformations, and the resulting gradient equivariance around the group orbit, gives information about how the loss gradient varies across parameter space (Fig. 1). To aggregate these gradients into an estimate of the loss curvature near $w$, we proceed as in most quasi-Newton methods (Nocedal and Wright, 2006; Li, 2017) whilst harnessing the network's symmetries. We assume the loss is twice differentiable and make a second-order Taylor approximation around $w^\star \equiv \mathcal{R}_1(w, \mathcal{G})$, the orbit average of $w$:

$$\begin{aligned}\mathcal{L}(w) &\approx \mathcal{L}(w^\star) + {g^\star}^\top(w - w^\star)\\ &\quad + \frac{1}{2}(w - w^\star)^\top H^\star(w - w^\star)\end{aligned} \tag{7}$$

where we have introduced the shorthand notation $g \equiv \nabla\mathcal{L}(w)$, $g^\star \equiv \nabla\mathcal{L}(w^\star)$, and $H^\star \equiv \nabla^2\mathcal{L}(w^\star)$. The accuracy of this approximation depends on the orbit size. In Lemma C.1 we show that the secant error is bounded by $\frac{M}{2}\|w - w^\star\|^2$ where $M$ is the Lipschitz constant of the Hessian, and that $\|w - w^\star\|$ grows monotonically with the group size. Reducing the orbit size will be crucial to recover Shampoo/Muon (Sec. 3.4). By differentiating with respect to $w$, the Taylor expansion implies the following secant condition:

$$g - g^\star \approx H^\star(w - w^\star) \tag{8a}$$

We assume that this approximation is valid along the entire orbit, i.e. $w \to Aw$, $g \to Ag$ for all transformations $A$ in the symmetry group. Furthermore, by Lemma 3.1, we have that $Aw^\star = w^\star$ and, by gradient equivariance, $Ag^\star = g^\star$. Therefore we have, for all $A$ in the symmetry group $\mathcal{G}$

$$A(g - g^\star) \approx H^\star A(w - w^\star) \tag{8b}$$

Multiplying Eq. 8b by its transpose, and performing an orbit average gives

$$S_g \approx H^\star S_w H^\star \tag{9a}$$

$$\text{with}\quad S_w \equiv \mathcal{R}_2(w - w^\star, w - w^\star, \mathcal{G}) \tag{9b}$$

$$\text{and}\quad S_g \equiv \mathcal{R}_2(g - g^\star, g - g^\star, \mathcal{G}). \tag{9c}$$

Most applications require positive semi-definite curvature approximations (Bottou et al., 2018). Beyond practical convenience, this is empirically well-motivated: the directional curvature along the gradient, $g^\top H g$, has been observed to be positive throughout most of training (Roulet et al., 2024), and the gradient itself tends to align with the top positive eigenvectors of the Hessian (Gur-Ari et al., 2018). This suggests that the positive semi-definite component of the Hessian plays a dominant role for gradient preconditioning, even though the full Hessian is generally indefinite. We show in Appendix B (see also Li, 2017) that the only positive semi-definite solution of Eq. 9a is

$$H^\star_{\mathrm{PD}} = S_w^{-\frac{1}{2}}\left(S_w^{\frac{1}{2}}S_g S_w^{\frac{1}{2}}\right)^{\frac{1}{2}}S_w^{-\frac{1}{2}}. \tag{10}$$

We show in Sec. 3.2 how to efficiently obtain the required orbit averages from $w$ and $g$ without having to compute any other transformed parameter nor loss gradient.

Empirically, we find that simplifying Eq. 10 by assuming $S_w$ to be approximately proportional to the identity provides a good approximation to the true curvature up to a scale factor (Sec. 4):

$$H^\star_g = S_g^{\frac{1}{2}}. \tag{11}$$

This is convenient for two reasons. First, $S_w$ may be rank-deficient, which would distort the curvature estimate upon inversion. Second, this simplification eliminates the need to estimate $S_w$, reducing computational overhead.

## 3.2. Structure of orbit averages

This section details how the invariance properties of orbit averages determine their structure. For a simple start, recall the example network of Fig. 1 (top) and its invariance to the simple permutation transformation discussed in Sec. 2.1. For the parameter tensor $C$, the second-order orbit average $S_{cc} \equiv \mathcal{R}_2(c, c, \mathcal{G})$ satisfies the equation $S_{cc} = (A_1 \otimes A_0)S_{cc}(A_1 \otimes A_0)^\top$, for any permutation matrix $A_1$, and with $A_0 = \mathrm{I}$ (c.f. Eq. 6b). In this case, the solution is $S_{mnop} = \delta_{mo}f^{(1)}_{np} + \mathbf{1}_{mo}f^{(2)}_{np}$ for some to-be-determined matrices of factors $\{f^{(1)}_{np}\}$ and $\{f^{(2)}_{np}\}$ (Bernacchia, 2025). This solution space is illustrated in Fig. 2 (top). Here, $S_{cc}$ has 32 factors to be determined.

Similarly, $\mathcal{R}_2(b, b, \mathcal{G})$ and $\mathcal{R}_2(b, c, \mathcal{G})$ obey analogous equations whose solutions can be found in Appendix D.5. Together, these orbit-averages give us the structure of each

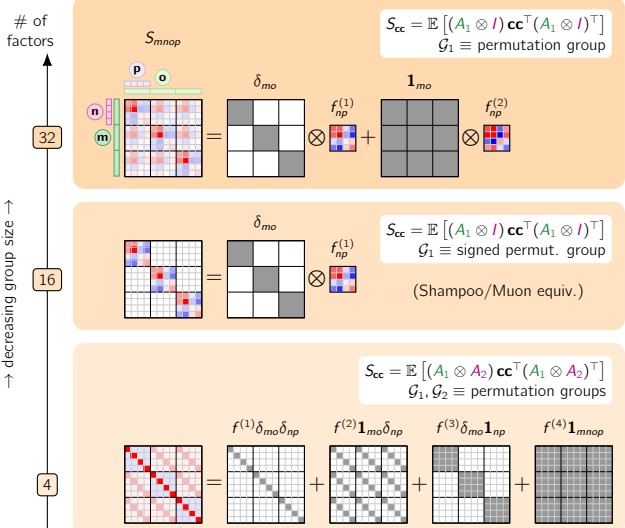

*Figure 2.* Structure of the second-order orbit average of $cc^\top$ with $c = \text{vec}(C)$, where $C$ is the $3 \times 4$ output matrix of the illustrative MLP of Fig. 1. In this case, $cc^\top$ has a natural order-4 tensor structure, and we denote the corresponding indices by $\{m, n, o, p\}$ (see diagram). Depending on the group considered (white insets), the resulting orbit average $S_{cc}$ can be very simple (bottom; four terms, each having only one scalar factor $f^{(\cdot)}$) or richer (top, two terms, but with matrix-valued factors $\{f_{np}^{(\cdot)}\}$ totaling 32 factors). Shampoo/Muon corresponds to the middle panel.

block of $S_w$ and $S_g$, as required for our Hessian approximations (Eqs. 10 and 11).

Now, suppose that the hidden layer had a `tanh` activation function. In this case, the network would still be invariant if $\mathcal{G}_1$ is the group of *signed* permutations (Chen et al., 1993). Averaging over the orbit of this larger group gives rise to more constrained solutions; for example, $\mathcal{R}_2(c, c)$ is now composed of a single basis element ($\delta_{mo}$), reducing the number of factors from 32 to 16 (Fig. 2, middle). This choice of group will give rise to the Shampoo optimizer – see Sec. 3.4.

Alternatively, suppose the MLP of Fig. 1 was used as an autoencoder. In this case, the loss function would be invariant not only to a permutation of the hidden layer, but also to the simultaneous permutation of the input and output layers. This gives orbit averages of the form $\mathcal{R}_2(c, c) = \mathbb{E}_{A_1, A_2}\left[(A_1 \otimes A_2)cc^\top(A_1 \otimes A_2)^\top\right]$ where the average is over two different permutation matrices $A_1$ and $A_2$. This further restricts the space of solutions, as shown in Fig. 2 (bottom), reducing the number of factors to only 4.

A key takeaway of this section is that orbit averages are linear superpositions of a small number of basis elements, weighted by factors (we will show how to determine these factors in the next section). The basis elements are sparse, binary tensors that afford a compact symbolic representation as Kronecker products of identity tensors and/or $\mathbf{1}$ tensors.

While the solutions are easy to intuit for simple cases, the more general case can be dealt with using a diagrammatic approach which we detail in Appendix D.5. This approach lets us find the structure of orbit averages of the form $S_{vv'} \equiv \mathcal{R}_2(v, v', \mathcal{G})$ where $v$ and $v'$ are tensors which are subject to a transformation group $\mathcal{G}$ as introduced in Sec. 2.1. In other words, we can enumerate all linearly independent solutions to the more general equation

$$S_{vv'} = \left(\bigotimes_{k=1}^{d} A_{i(k)}\right) S_{vv'} \left(\bigotimes_{k=1}^{d'} A_{i'(k)}\right)^\top \tag{12}$$

that must hold for any $\{A_1, A_2, \ldots\}$ in the group.

As an example application of this general theory, Fig. 3 illustrates the $S_g$ of a single-layer Transformer, which comprises six different parameter tensors: embedding weights $W_{\text{emb}}$; attention weights $W_{QK}, W_V$ and associated projection $W_{\text{proj}}$; and feed-forward weights $W_{\text{ff}_1}$ and $W_{\text{ff}_2}$. The structured group for which the orbit average is computed is detailed in Appendix I. This example shows that even after orbit averaging, $S_g$ retains a lot of structure.

### 3.3. Computing the factors

We have shown that the curvature can be represented in terms of a small number of factors, that depend on the symmetry group. Here we show how to efficiently compute these factors. We prove the following Lemma in Appendix E:

**Lemma 3.2.** *The orbit average is the orthogonal projection onto the space of invariants. Consequently, the first-order orbit average of $v$ denoted by $s(f^\star) \equiv \mathcal{R}_1(v, \mathcal{G})$ is the invariant that is nearest to $v$ by the square norm:*

$$f^\star = \arg\min_f |s(f) - v|^2. \tag{13a}$$

*Similarly, for the second-order orbit average of $vv^\top$ denoted by $S(f^\star) \equiv \mathcal{R}_2(v, v', \mathcal{G})$,*

$$f^\star = \arg\min_f \left|S(f) - vv'^\top\right|_F^2. \tag{13b}$$

This lemma implies that we can find the factors $f$, associated with each basis function (c.f. Sec. 3.2) by least-squares, which can be solved analytically in most cases.

As an example, we consider again the case in which the invariant is equal to $S_{mnop} = \delta_{mo}f_{np}$ (Sec. 3.2; Fig. 2, middle). This example will be instrumental for deriving the equivalence with Shampoo and Muon in Sec. 3.4. In matrix notation, the invariant is written as $S(F) = F \otimes I$, where $F$ is the matrix of factors $f_{np}$. Then, denoting by $G$ the gradient matrix of the corresponding tensor, the solution of Eq. 13b with $v = v' = g = \text{vec}(G)$ is equal to

$$\mathcal{R}_2(g, g, \mathcal{G}) = GG^\top \otimes I. \tag{14}$$

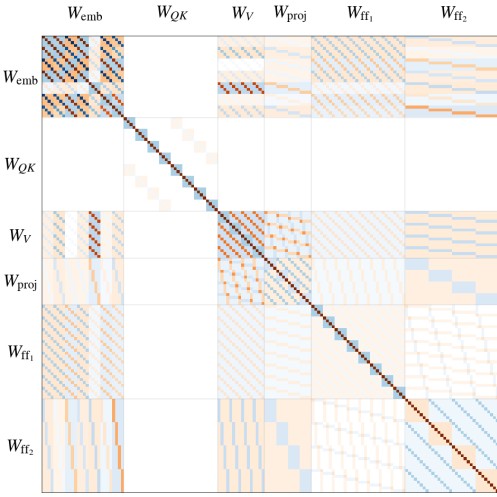

*Figure 3.* Orbit averaged gradient $\mathcal{R}_2(\boldsymbol{g}, \boldsymbol{g}, \mathcal{G})$ of a Transformer layer, normalized to unit diagonal. Each block of the matrix consists of a handful of colors, reflecting the small number of bases and associated factors $\boldsymbol{f}$.

More generally, we provide a PyTorch-based library[1] that analyses user-provided symbolic specifications of a network's symmetries (see Appendix M for example code), and just-in-time compiles them down to efficient routines for computing orbit averages such as $S_{\boldsymbol{w}}$ and $S_{\boldsymbol{g}}$. Importantly, any matrix analytic function (including e.g. their inverse square root, useful below) has the same structure, with factors that can be computed efficiently through simple model reduction techniques (Appendix L in Bernacchia, 2025).

### 3.4. Orbit averaging reduces to Shampoo/Muon under specific symmetry group choices

Here, we use the symmetry-based Hessian approximation of Eq. 11 to construct the 'Symo' optimizer (named for its exploitation the weight-space symmetries, Bernacchia, 2025), which is defined by the update

$$\boldsymbol{w}_{t+1} = \boldsymbol{w}_t - \eta \big(H_{\boldsymbol{g}}^{\star}\big)^{-1} \boldsymbol{g}_t, \qquad (15)$$

where $H_{\boldsymbol{g}}^{\star}$ is defined in Eq. 11 (see Appendix G for practical details of training). The following Lemma shows that a specific restriction of the symmetry group recovers the Shampoo and Muon updates (proof in Appendix H).

**Lemma 3.3** (Symo reduces to Shampoo/Muon under Kronecker block-diagonal curvature). *Let $W^{(i)} \in \mathbb{R}^{n \times m}$ denote the $i$-th parameter tensor of a model, and let the curvature matrix $S_{\boldsymbol{g}}$ be block diagonal, with the $i$-th diagonal block $S_{\boldsymbol{g}}^{(ii)}$ corresponding to $\boldsymbol{w}^{(i)} = vec(W^{(i)})$. We denote by $G^{(i)} \in \mathbb{R}^{n \times m}$ the gradient of the loss with respect to $W^{(i)}$. Suppose each block takes the form*

$$S_{\boldsymbol{g}}^{(ii)} = I \otimes F \quad or \quad S_{\boldsymbol{g}}^{(ii)} = F \otimes I, \qquad (16)$$

[1] https://github.com/mtkresearch/symm_opt

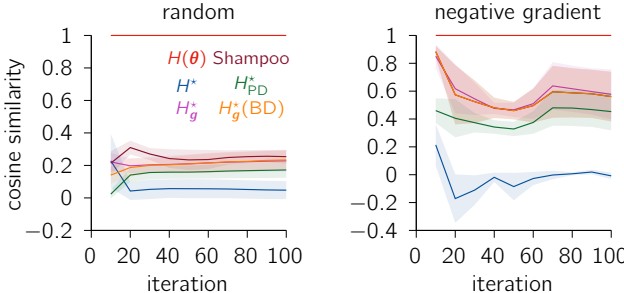

*Figure 4.* Hessian approximation experiments. Cosine similarities between $\boldsymbol{g}' - \boldsymbol{g}$ and $\hat{H}(\boldsymbol{w}' - \boldsymbol{w})$, averaged across random directions (left) and in the negative gradient direction (right); with $\pm 95\%$ confidence intervals over 5 random seeds. See main text for details. Note that in the gradient direction, Shampoo and $H_{\boldsymbol{g}}^{\star}$ (BD) overlap completely (consistent with the mathematical equivalence we proved).

*where $F$ is a matrix of factors. Then the Symo update to $W^{(i)}$ reduces to the following:*

$$\begin{cases} \sqrt{n}\, G^{(i)} \big((G^{(i)})^{\top} G^{(i)}\big)^{-\frac{1}{2}}, & when\ S_{\boldsymbol{g}}^{(ii)} = I \otimes F \\ \sqrt{m}\, \big(G^{(i)}(G^{(i)})^{\top}\big)^{-\frac{1}{2}} G^{(i)}, & when\ S_{\boldsymbol{g}}^{(ii)} = F \otimes I \end{cases}$$

*Hence, the Symo update recovers the Shampoo/Muon update (up to a constant scaling factor that corresponds to whitened Shampoo (Gong et al., 2025); Appendix H).*

For both an MLP (Appendix H.2) and a Transformer (Appendix H.3), we can choose symmetry groups and network architectures such that Eq. 16 holds. For an MLP with an even number of layers, we assign the trivial (identity) group to every even-indexed layer, while odd-indexed layers are associated with distinct signed permutation groups. For the Transformer, we assign the identity group to all layers whose representations have the embedding dimension, which most architecture will suffice with residual connections. Note that restricting any group to identity effectively reduces the size of the orbit, and arguably improves the approximation in Eq. 7. From Symo's perspective, Muon and Shampoo do not fully exploit the available symmetries; instead, they assume identity symmetries for certain layers, striking a balance between approximation quality and cost.

## 4. Evaluation of the Hessian approximation

We evaluate the quality of our proposed approximations of the Hessian, Eqs. 10 and 11, and compare them with other Hessian approximations. The comparison is based on the secant condition of Eq. 8b: we compute the cosine similarity between $\boldsymbol{g} - \boldsymbol{g}'$ and $\hat{H}(\boldsymbol{w}' - \boldsymbol{w})$, where $\boldsymbol{w}$ denotes the parameter vector at a given point during optimization, $\boldsymbol{w}' = \boldsymbol{w} + \Delta\boldsymbol{w}$, $\boldsymbol{g} = \nabla\mathcal{L}(\boldsymbol{w})$, and $\boldsymbol{g}' = \nabla\mathcal{L}(\boldsymbol{w}')$. For small $\Delta\boldsymbol{w}$, using the exact Hessian gives a similarity equal to one. We consider two choices for $\Delta\boldsymbol{w}$: (1) taking a step of size $r$ in a random direction and average the cosine similarity across

$N$ different random directions; (2) taking a step of size $r$ in the negative gradient direction $-\boldsymbol{g}$. We measure similarities at different points along the optimization trajectory (see Appendix J for details).

We compare against the true Hessian at $\boldsymbol{w}$ ($H = \nabla^2 \mathcal{L}(\boldsymbol{w})$) and five other Hessian approximations: (i) the true Hessian evaluated at the orbit-averaged parameters, $H^\star$ (Eq. 7); (ii) $H^\star_{\mathrm{PD}}$ (Eq. 10); (iii) $H^\star_{\boldsymbol{g}}$ (Eq. 11); (iv) block-diagonal $H^\star_{\boldsymbol{g}}$ (denoted as $H^\star_{\boldsymbol{g}}(\mathrm{BD})$); (v) the Shampoo Hessian approximation (Sec. 2.3). Since we would like to compute exact Hessians for comparison, we consider a small four-layer MLP model in a teacher-student setting (mapping the input $x \in \mathbb{R}^{100}$ through layer widths $70 \to 70 \to 70 \to 40$; $\mathtt{tanh}$ activation function; $D = 19{,}850$ parameters in total; see Appendix J for full details). To obtain mathematical equivalence between $H^\star_{\boldsymbol{g}}(\mathrm{BD})$ and Shampoo's curvature matrix, we assign the trivial (identity) group to the input, output and middle layers, and a distinct signed permutation group to the first and last layer (see Sec. 3.4). We note that, while $H^\star_{\boldsymbol{g}}(\mathrm{BD})$ is equivalent to Shampoo in terms of the optimization *update*, their Hessian approximations coincide only up to a scaling factor and when Eq. 101 is satisfied (see Appendix H).

In random directions, Shampoo exhibits the highest similarity among all approximations (slightly above 0.2) (Fig. 4, left). All three of our Hessian approximations – $H^\star_{\mathrm{PD}}$, $H^\star_{\boldsymbol{g}}$, and $H^\star_{\boldsymbol{g}}(\mathrm{BD})$ exhibit comparable Hessian approximation quality (between 0.1 and 0.2). While $H^\star$ has near zero cosine similarity. We note that similarities of 0.1 - 0.2 are quite high considering the high dimensionality of the parameter space, since the average cosine similarity between two random unit vectors is approximately $0.007 \approx 1/\sqrt{D}$.

Along the true gradient direction, which is more relevant for optimization, all three of our Hessian approximations achieve consistently higher similarity across training iterations (Fig. 4, right). Firstly, $H^\star_{\boldsymbol{g}}$ achieving better approximation to the Hessian compared to $H^\star_{\mathrm{PD}}$, despite being an approximation to the latter, is due to the rank deficiency of $S_{\boldsymbol{w}}$ in Eq. 10 (see Appendix F for empirical verification). Inversion of $S_{\boldsymbol{w}}$ amplifies noise along singular directions which makes $H^\star_{\mathrm{PD}}$ less stable than $H^\star_{\boldsymbol{g}}$, providing strong motivation for using the approximation $S_{\boldsymbol{w}} \sim I$ beyond computational savings. Secondly, restricting $H^\star_{\boldsymbol{g}}$ to be block-diagonal ($H^\star_{\boldsymbol{g}}(\mathrm{BD})$) does not degrade curvature estimation, indicating that off-diagonal contributions to curvature are small and we can gain further computational savings by using a block-diagonal approximation. In contrast, $H^\star$ approximates the Hessian very poorly in this direction, evident in its consistently low (nearly 0) and occasionally negative cosine similarity with the Hessian. Finally, we demonstrate the equivalence between $H^\star_{\boldsymbol{g}}(\mathrm{BD})$ and Shampoo (underneath the $H^\star_{\boldsymbol{g}}(\mathrm{BD})$ curve), as their cosine similarity curves

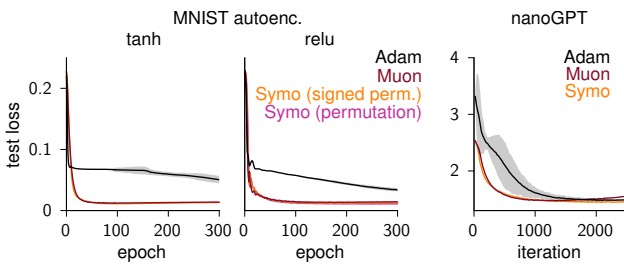

*Figure 5.* We demonstrate the equivalence of Symo and Muon on both the MNIST autoencoder task and the nanoGPT next-character prediction task. For comparison, we also train networks using Adam, showing the superior performance of Symo and Muon relative to Adam. **Left**: Test mean-squared error (MSE) for the MNIST autoencoder under two activation functions. **Right**: Test loss for nanoGPT. Note that the **Symo** and **Muon** curves overlap.

overlap along the negative gradient direction. These observations provide a solid basis for interpreting the effectiveness of our approach in capturing curvature along optimization-relevant directions.

## 5. Empirical equivalence with Shampoo/Muon

In Sec. 3.4 and Sec. 4, we provided both theoretical and empirical justifications for the equivalence between Symo and Shampoo/Muon updates when restricting group symmetries. In this section, we demonstrate their equivalence in more practical settings. In particular, we show that Symo's training curves closely match those of Muon, even in scenarios where the two optimizers are not mathematically identical due to differences in scaling and training heuristics. We illustrate this *empirical* equivalence with two tasks: an MNIST deep autoencoder with an MLP, and a character-level text prediction task with nanoGPT.

### 5.1. MNIST autoencoder

We train a deep autoencoder on the MNIST dataset (Le-Cun et al., 1998; Martens and Grosse, 2015; Goldfarb et al., 2020). The model is a fully-connected symmetric autoencoder with affine layers (including biases) and $\mathtt{tanh}$ non-linearities, mapping the input $x \in \mathbb{R}^{784}$ through layers of sizes $784 \to 1000 \to 500 \to 250 \to 30 \to 250 \to 500 \to 1000 \to 784$, with a $\mathtt{sigmoid}$ activation at the output layer. The network has 2.8M learnable parameters (details of experimental setup and hyperparameters in Appendices K.1 and K.3).

The conditions for exact mathematical equivalence between Symo and Muon for this autoencoder task are listed in List K.1.2. Importantly, we only satisfy the following: Item 1, corresponding to restricting the Symo preconditioner to be block-diagonal; Item 2, corresponding to applying identity symmetry for the input layer and alternating between signed permutation symmetries and identity sym-

metries for subsequent layers; and Item 7, corresponding to assuming the orbit average of the gradients to be zero (Appendix K.1). Empirically, the equivalence appears relatively insensitive to the specific choice of nonlinear activation; for example, **Symo** and Muon behave near-identically on MNIST with ReLU activation, and in nanoGPT with GELU activation, neither of which lead to exact signed permutation invariance.

**Symo** and Muon both outperform Adam, and the near perfect overlap in their respective training curves empirically demonstrates their equivalence (Fig. 5, left). We attribute the very small differences between **Symo** and Muon curves as arising from Item 3, Item 4, Item 5, and Item 6 not being satisfied. We deliberately chose not to satisfy these conditions: Item 3 would break the mathematical soundness of Symo, whereas Item 4, Item 5, and Item 6 are Muon-specific training heuristics that improve convergence speed and training efficiency (Jordan et al., 2024).

### 5.2. NanoGPT

Next, we train a transformer (nanoGPT; Karpathy, 2026) to perform a character-level prediction task on the Shakespeare dataset (Karpathy, 2015). The model consists of 6 transformer layers with 6 attention heads per layer and a hidden embedding dimension of 384. The network has approximately 10 million learnable parameters (see details of experimental setup in Appendix K.2 and hyperparameters in Appendix K.3). This compact architecture serves as a lightweight benchmark for evaluating optimization methods on sequence modeling tasks.

The conditions for exact mathematical equivalence between Symo and Muon for the nanoGPT task are listed in List K.2.2. We satisfy only Item 1, Item 2, and Item 7, which are the same three conditions satisfied in the autoencoder setting, as well as Item 6, corresponding to the use of Adam for training specific subsets of weights (see Appendix K.2).

**Symo** and Muon again produce closely matching training and evaluation curves, while outperforming Adam (Fig. 5, right). Any small differences between them arise from Item 3, Item 4, Item 5 not being satisfied. We chose not to satisfy these conditions for the same reasons as explained in Sec. 5.1. Together, the autoencoder and nanoGPT experiments demonstrate that **Symo** and Muon can exhibit practical equivalence in real-world settings, while preserving the mathematical soundness of Symo and the effective training heuristics of Muon.

## 6. Exploring choices of symmetry groups

In Fig. 6, we show how different choices of symmetry group – which all leave the loss invariant – affect the performance of our symmetry-based optimizer. We train deep (ReLU)

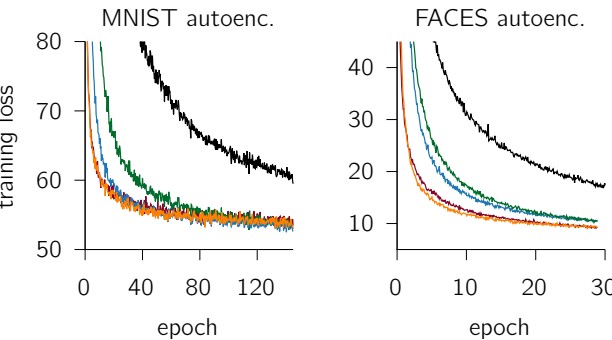

*Figure 6.* Exploring various choices of symmetry groups on autoencoder optimization benchmarks. See Goldfarb et al. (2020) for details of datasets and comparisons to other popular 2nd-order optimizers. For **Symo**, we considered the invariant action of group where every layer within the autoencoder can be permuted, except for the input and output layers. In **Symo (larger group)**, this group is enlarged to also include a simultaneous permutation of the input and output layers (which also leaves the autoencoding loss invariant). In **Symo (smaller group)**, we restrict the group such that only every other layer in the autoencoder is permuted, and restrict the Hessian approximation to block-diagonal form. Theoretically, this gives approximate equivalence to **Shampoo** (c.f. Sec. 3.4), which is corroborated numerically here.

MLPs on a standard autoencoders benchmark (Goldfarb et al., 2020), and compare Adam, Shampoo, and Symo with various choices of symmetry groups, with learning rates optimized through grid search (Appendix L). The largest group to which deep MLPs are universally invariant is one in which each internal layer is permuted, but where the input and output layers are pinned (referred to as simply "**Symo**"). This optimizer is already much closer to Shampoo than to Adam. Interestingly, even with the largest possible group allowed by the autoencoding loss – whereby all layers, including inputs and outputs, are permuted, resulting in a Hessian approximation with only 64 free parameters – **Symo** (larger group) still outperforms Adam by a large margin. Conversely, by allowing permutations only for every odd layer and discarding off-diagonal Hessian blocks (**Symo** (smaller group)), the expressiveness of the Hessian approximation becomes similar to Shampoo's and yields comparable optimization performance. This suggests that this particular choice of symmetry group hits a sweet spot in terms of tractability and expressiveness.

## 7. Summary and limitations

This work introduced a new framework for approximating the curvature of large neural networks by harnessing weight-space symmetries. We demonstrated that by averaging over the orbit generated by a single gradient, we

can obtain useful curvature information which can then be used for optimization, recovering Shampoo/Muon as special cases. Our approach could inspire novel curvature approximations by considering symmetries inherent to different model classes. Beyond optimization, future work may extend this framework to applications like Bayesian inference, continual learning, and model compression.

Nevertheless, the framework has certain limitations. The accuracy of the second-order Taylor expansion around the orbit average (Eq. 7) remains unclear. We showed that reducing the size of the symmetry group, and therefore the orbit size, recovered well known optimizers, such as Shampoo and Muon. We provide a preliminary bound on the Taylor approximation error in terms of the orbit radius (Lemma C.1) confirming that smaller groups yield tighter approximations. However, bounding the orbit radius for *trained* (non-random) weights, and understanding how the Hessian Lipschitz constant $M$ interacts with architecture and training dynamics, remain open questions for future work.

We note that our framework provides a principled route from symmetries to structured curvature approximations, and we show empirically that these approximations correlate well with the true Hessian along the gradient direction (see Fig. 4). However, we do not establish a formal guarantee that Shampoo/Muon's preconditioner shares eigenvectors with the Hessian, nor do we prove that the gradient outer product $gg^\top$ converges to the Hessian in any formal sense. Establishing such a connection rigorously remains an open problem – one that, to our knowledge, no existing work has resolved either.

Additionally, as with many curvature approximations, we rely in practice on a block-diagonal Hessian due to memory constraints – for example, in the nanoGPT task (Sec. 5.2) – which leaves cross-tensor interactions unexplored. The possibility of improving curvature approximations by including off-diagonal blocks, in a scalable way, may be the subject of further investigation.

In terms of practical significance, although we have automated the analytical solutions for invariance equations in Eq. 12 under any combination of common groups, meaningful comparison of computational and memory cost with other optimizers would still require more engineering time to ensure parallel computations in face of highly heterogeneous group structures. Currently, users need only specify the group settings, as shown in Appendix M, and our library automates the derivation of orbit averages. However, group selections for new architectures remains an open question, and the full automation based on just the computational graph of any architectures are left for future work.

## Impact statement

This paper presents work whose goal is to advance the field of machine learning. There are many potential societal consequences of our work, none of which we feel must be specifically highlighted here.

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

# A. Proof of Lemma 3.1

Let $A^{(i)} \in \mathcal{G}$ be any fixed group element. Since $\mathcal{R}_1$ is linear in $\boldsymbol{v}$, it can be represented by a matrix $P$:

$$\mathcal{R}_1(\boldsymbol{v}, \mathcal{G}) = P\boldsymbol{v} \quad \text{with} \quad P \equiv \frac{1}{|\mathcal{G}|} \sum_{j=1}^{|\mathcal{G}|} A^{(j)}, \tag{17}$$

where the index $j$ runs over all elements of $\mathcal{G}$, and we recall that $\mathcal{G} = \mathcal{G}_0 \times \mathcal{G}_1 \times \ldots \mathcal{G}_m$ is the direct product of groups acting on the individual tensors, and is itself a group.

For fixed $i$, the map $j \mapsto k$ defined by $A^{(i)} A^{(j)} = A^{(k)}$ is a bijection on $\mathcal{G}$: it is injective since $A^{(i)} A^{(j_1)} = A^{(i)} A^{(j_2)}$ implies $A^{(j_1)} = A^{(j_2)}$ by left-cancellation, and surjectivity follows since any $A^{(k)}$ is the image of $A^{(j)} = (A^{(i)})^{-1} A^{(k)}$. Therefore, summing over $k$ is equivalent to summing over $j$:

$$A^{(i)} \mathcal{R}_1(\boldsymbol{v}, \mathcal{G}) = A^{(i)} P\boldsymbol{v} = \frac{1}{|\mathcal{G}|} \sum_{j=1}^{|\mathcal{G}|} A^{(i)} A^{(j)} \boldsymbol{v} = \frac{1}{|\mathcal{G}|} \sum_{k=1}^{|\mathcal{G}|} A^{(k)} \boldsymbol{v} = P\boldsymbol{v} = \mathcal{R}_1(\boldsymbol{v}, \mathcal{G}). \tag{18}$$

For $\mathcal{R}_2$, let $B \equiv \bigotimes_{k=1}^{d'} A_{i'(k)}$ denote the representation acting on $\boldsymbol{v}'$. The same rearrangement argument applied simultaneously to left-multiplication by $A^{(i)}$ and right-multiplication by $B^{(i)^\top}$ gives

$$A^{(i)} \mathcal{R}_2(\boldsymbol{v}, \boldsymbol{v}', \mathcal{G}) B^{(i)^\top} = \frac{1}{|\mathcal{G}|} \sum_{j=1}^{|\mathcal{G}|} A^{(i)} A^{(j)} \boldsymbol{v} \boldsymbol{v}'^\top B^{(j)^\top} B^{(i)^\top}$$

$$= \frac{1}{|\mathcal{G}|} \sum_{k=1}^{|\mathcal{G}|} A^{(k)} \boldsymbol{v} \boldsymbol{v}'^\top B^{(k)^\top} = \mathcal{R}_2(\boldsymbol{v}, \boldsymbol{v}', \mathcal{G}), \tag{19}$$

where we used the fact that $A^{(i)} A^{(j)} = A^{(k)}$ implies $B^{(j)} B^{(i)} = B^{(k)}$ (and thus $B^{(j)^\top} B^{(i)^\top} = B^{(k)^\top}$), since both $A$ and $B$ are representations of the same group element.

When $\mathcal{G}$ contains continuous compact groups (such as orthogonal groups), the finite sum $\frac{1}{|\mathcal{G}|} \sum_j (\cdot)$ is replaced by integration against the Haar measure $\mu$, which exists and is unique up to normalization for compact groups. The analogous rearrangement step is justified by left-invariance of the Haar measure, $\int_\mathcal{G} f(g) \, d\mu(g) = \int_\mathcal{G} f(hg) \, d\mu(g)$ for any fixed $h \in \mathcal{G}$. Hence the Lemma extends to continuous compact groups.

# B. Proof of formula (10) for the Hessian

We rewrite Eq. 9a

$$S_{\boldsymbol{g}} = H^\star S_{\boldsymbol{w}} H^\star. \tag{20}$$

We note that, by definition, $S_{\boldsymbol{w}}$ and $S_{\boldsymbol{g}}$ are positive semidefinite. Multiplying left and right of Eq. 20 by $S_{\boldsymbol{w}}^{\frac{1}{2}}$ (defined as the symmetric square root) we obtain

$$S_{\boldsymbol{w}}^{\frac{1}{2}} S_{\boldsymbol{g}} S_{\boldsymbol{w}}^{\frac{1}{2}} = \left( S_{\boldsymbol{w}}^{\frac{1}{2}} H^* S_{\boldsymbol{w}}^{\frac{1}{2}} \right)^2. \tag{21}$$

We note that the left hand side is positive semidefinite. We denote its eigenvalue decomposition as

$$S_{\boldsymbol{w}}^{\frac{1}{2}} S_{\boldsymbol{g}} S_{\boldsymbol{w}}^{\frac{1}{2}} = V D^2 V^\top \tag{22}$$

where columns of $V$ and the diagonal of $D^2$ are, respectively, its orthogonal eigenvectors and eigenvalues. Then, assuming that $S_{\boldsymbol{w}}$ is not singular, the general solution of Eq. 21 is

$$H^* = S_{\boldsymbol{w}}^{-\frac{1}{2}} V (\pm D) V^\top S_{\boldsymbol{w}}^{-\frac{1}{2}}. \tag{23}$$

The expression $(\pm D)$ denotes the $2^p$ possible diagonal matrices with arbitrary signs for its diagonal entries. By Sylvester's law of inertia, since $H^*$ is a congruent transform of $(\pm D)$, it has the same number of positive and negative eigenvalues. Therefore, the number of positive and negative eigenvalues of $H^*$ is arbitrary and the only positive definite solution is

$$H_{\text{PD}}^\star = S_{\boldsymbol{w}}^{-\frac{1}{2}} \left( S_{\boldsymbol{w}}^{\frac{1}{2}} S_{\boldsymbol{g}} S_{\boldsymbol{w}}^{\frac{1}{2}} \right)^{\frac{1}{2}} S_{\boldsymbol{w}}^{-\frac{1}{2}}. \tag{24}$$

## C. Taylor approximation quality

**Lemma C.1** (Orbit distance and Taylor approximation quality). *Let $L : \mathbb{R}^d \to \mathbb{R}$ be twice continuously differentiable with $M$-Lipschitz continuous Hessian, i.e. $\|\nabla^2 L(u) - \nabla^2 L(v)\| \le M\|u - v\|$ for all $u, v$. Let $\mathcal{G}$ be an orthogonal symmetry group under which $L$ is invariant, and let $P \equiv \mathcal{R}_1(\cdot, \mathcal{G})$ denote the orthogonal projection onto the $\mathcal{G}$-invariant subspace $V \subseteq \mathbb{R}^d$ (Lemma 3.2). Define $w^\star \equiv Pw$. Then:*

*(i)* ***Equidistance.*** *Every point on the orbit is equidistant from $w^\star$:*

$$\|Aw - w^\star\| = \|w - w^\star\|, \quad \forall A \in \mathcal{G}. \tag{25}$$

*(ii)* ***Secant error bound.*** *The error in the secant condition, Eq. 8b, satisfies*

$$\|\nabla L(Aw) - g^\star - H^\star(Aw - w^\star)\| \le \frac{M}{2}\|w - w^\star\|^2, \quad \forall A \in \mathcal{G}, \tag{26}$$

*where $g^\star \equiv \nabla L(w^\star)$ and $H^\star \equiv \nabla^2 L(w^\star)$.*

*(iii)* ***Expected orbit radius.*** *If $w \sim \mathcal{N}(0, \sigma^2 I_d)$, then*

$$\frac{\mathbb{E}\|w - w^\star\|^2}{\mathbb{E}\|w\|^2} = 1 - \frac{\dim(V)}{d}. \tag{27}$$

*In particular, restricting to a smaller group $\mathcal{G}' \subset \mathcal{G}$ enlarges the invariant subspace ($\dim(V') \ge \dim(V)$), reduces $\|w - w^\star\|$, and tightens the secant error bound.*

*Proof.* **(i)** By Lemma 3.1, $Aw^\star = w^\star$ for all $A \in \mathcal{G}$. Since $A$ is orthogonal,

$$\|Aw - w^\star\|^2 = \|A(w - w^\star)\|^2 = (w - w^\star)^\top A^\top A(w - w^\star) = \|w - w^\star\|^2. \tag{28}$$

**(ii).** Since $L$ has $M$-Lipschitz continuous Hessian, the standard Taylor remainder bound (Nesterov et al., 2018, Lemma 1.2.4) for any $u$ in the domain gives,

$$\|\nabla L(u) - \nabla L(w^\star) - H^\star(u - w^\star)\| \le \frac{M}{2}\|u - w^\star\|^2. \tag{29}$$

Substituting $u = Aw$ and using gradient equivariance $\nabla L(Aw) = A\nabla L(w)$ together with Lemma 3.1, i.e. $Aw^\star = w^\star$ and $Ag^\star = g^\star$, the left-hand side becomes $\|A(g - g^\star) - H^\star A(w - w^\star)\|$. The right-hand side equals $\frac{M}{2}\|Aw - w^\star\|^2 = \frac{M}{2}\|w - w^\star\|^2$ by (i).

**(iii).** Since $P$ is an orthogonal projection (Lemma 3.2), $\|w - w^\star\|^2 = \|(I - P)w\|^2 = \|w\|^2 - \|Pw\|^2$. For $w \sim \mathcal{N}(0, \sigma^2 I_d)$, $Pw$ is the projection of an isotropic Gaussian onto a $\dim(V)$-dimensional subspace, so $\mathbb{E}\|Pw\|^2 = \text{tr}(P\mathbb{E}[ww^\top]) = \sigma^2 \dim(V)$ and $\mathbb{E}\|w\|^2 = \sigma^2 d$. Hence

$$\frac{\mathbb{E}\|w - w^\star\|^2}{\mathbb{E}\|w\|^2} = \frac{\sigma^2(d - \dim(V))}{\sigma^2 d} = 1 - \frac{\dim(V)}{d}. \tag{30}$$

Finally, if $\mathcal{G}' \subset \mathcal{G}$, then $V \subseteq V'$ meaning that a smaller group imposes fewer constraints and therefore more vectors are invariant giving $\dim(V') \ge \dim(V)$, a smaller expected orbit radius, and therefore a tighter secant error bound. □

## D. Algebraic Structure of Network Symmetries

### D.1. Definitions and Notation

We formalize the algebraic structures used to describe the geometry of the weight space. Let $\mathcal{V}$ be a finite-dimensional vector space over $\mathbb{R}$.

**Definition D.1** (Automorphism Group.). The automorphism group of $\mathcal{V}$, denoted $\mathrm{Aut}(\mathcal{V})$, is the set of all invertible linear transformations from $\mathcal{V}$ to itself. This coincides with the general linear group $\mathrm{GL}(\mathcal{V}) \cong \mathrm{GL}_n(\mathbb{R})$ where $n = \dim(\mathcal{V})$. The group operation is function composition, i.e. matrix multiplication.

**Definition D.2** (Direct Product of Groups.). Let $\mathcal{G}_1, \ldots, \mathcal{G}_k$ be a collection of groups. Their direct product $\mathcal{G} = \mathcal{G}_1 \times \cdots \times \mathcal{G}_k = \prod_{i=1}^{k} \mathcal{G}_i$ is the set of tuples $(g_1, \ldots, g_k)$ with component-wise operations. If $g = (g_1, \ldots, g_k)$ and $h = (h_1, \ldots, h_k)$, then $g \cdot h = (g_1 h_1, \ldots, g_k h_k)$. The order of the product group is the product of the orders of the component groups (Lang, 2012).

**Definition D.3** (Direct Sum of Representations.). Let $\rho_1 : \mathcal{G} \to \mathrm{GL}(V_1)$ and $\rho_2 : \mathcal{G} \to \mathrm{GL}(V_2)$ be representations of a group $\mathcal{G}$. The direct sum representation $\rho = \rho_1 \oplus \rho_2$ acts on the vector space $V_1 \oplus V_2$. The matrix representation of this action is block-diagonal:

$$(\rho_1 \oplus \rho_2)(a) = \begin{pmatrix} \rho_1(a) & 0 \\ 0 & \rho_2(a) \end{pmatrix}. \tag{31}$$

This construction allows the simultaneous analysis of independent actions on disjoint subspaces (Fulton and Harris, 2013).

**Definition D.4** (Commutant Algebra). Let $V$ be a vector space. The commutant algebra is simply the collection of all linear operators $T$ that commute with every transformation in the group $\mathcal{G}$, such that:

$$T\rho(a) = \rho(a)T, \quad \text{for all } a \in \mathcal{G}. \tag{32}$$

Note that Definition D.4 is also often called *centralizer algebra*. Commutant hightlights algebraic property, whereas centralizer focuses on geometric stability, i.e. $T$ is central to the group $\mathcal{G}$.

### D.2. Derivation of the Kronecker Representation

We demonstrate the origin of the tensor product structure in Eq. Eq. 43 using a deep linear network. Consider a network with $L$ layers defined by the composition of weight matrices $\{W_l\}_{l=0}^{L-1}$ where $W_l \in \mathbb{R}^{d_{l+1} \times d_l}$. The output $\mathbf{y}$ for an input $\mathbf{x}$ is:

$$\mathbf{y} = W_{L-1} W_{L-2} \ldots W_0 \mathbf{x}. \tag{33}$$

We introduce a set of invertible transformations defined by the abstract group elements $\{g_l\}_{l=0}^{L}$, where $g_l$ belongs to the symmetry group $\mathcal{G}_l$ of layer $l$. Let $G_l \in \mathrm{GL}(d_l)$ denote the matrix representation of the element $g_l$ acting on the vector space $\mathbb{R}^{d_l}$, formally $G_l \equiv \rho_l(g_l)$. Inserting the identity operator $I = G_l^{-1} G_l$ between adjacent layers preserves the functional mapping:

$$\mathbf{y} = W_{L-1}(A_{L-1}^{-1} A_{L-1}) W_{L-2} \ldots (A_1^{-1} A_1) W_0 \mathbf{x} \tag{34}$$
$$= (W_{L-1} A_{L-1}^{-1})(A_{L-1} W_{L-2} A_{L-2}^{-1}) \ldots (A_1 W_0) \mathbf{x}. \tag{35}$$

To maintain strict invariance of the output $\mathbf{y}$ with respect to the input $\mathbf{x}$ (assuming $G_0 = I$ and $G_L = I$ or absorbing external transforms into the data), the weights must transform according to the rule:

$$W_l' = A_{l+1} W_l A_l^{-1}. \tag{36}$$

We vectorize the parameters to define the action on the total parameter vector $\boldsymbol{w}$. We utilize the vectorization identity $\mathrm{vec}(AXB) = (B^\top \otimes A)\mathrm{vec}(X)$. Applying this to Eq. (36):

$$\mathrm{vec}(W_l') = \mathrm{vec}(A_{l+1} W_l A_l^{-1}) = (A_l^{-T} \otimes A_{l+1})\mathrm{vec}(W_l). \tag{37}$$

Restricting our analysis to orthogonal groups where $A^{-1} = A^\top$, the transformation simplifies to:

$$\mathrm{vec}(W_l') = (A_l \otimes A_{l+1})\mathrm{vec}(W_l). \tag{38}$$

This confirms that the action on a single weight matrix is the tensor product of the actions on its input and output spaces. Stacking the vectorized weights of all layers $\boldsymbol{w} = [\mathrm{vec}(W_0)^\top, \ldots, \mathrm{vec}(W_{L-1})^\top]^\top$, the global group action $\rho(a)$ becomes the direct sum of these Kronecker products:

$$\rho(a)\boldsymbol{w} = \left[ \bigoplus_{l=0}^{L-1} (A_l \otimes A_{l+1}) \right] \boldsymbol{w}. \tag{39}$$

This derivation justifies the block-diagonal Kronecker structure presented in Eq. (43).

**D.3. Formal Group representation of neural networks**

In Sec. 2.1, we defined that neural networks as a composition of linear maps $\{W_l\}_{l=0}^{L-1}$ interleaved with non-linearities. The weights for each layer $W_l$ live in the space of linear maps $\text{Hom}(\mathcal{V}_l, \mathcal{V}_{l+1})$. Therefore, we can write the global vectorized parameter vector space $\mathcal{W}$ of the whole network as the *direct sum* of individual layer's vector spaces:

$$\mathcal{W} = \bigoplus_{l=0}^{L-1} \text{Hom}(\mathcal{V}_l, \mathcal{V}_{l+1}) \cong \bigoplus_{l=0}^{L-1} (\mathcal{V}_{l+1} \otimes \mathcal{V}_l^*). \tag{40}$$

In fact, when network consists of weights which are tensors of order higher than 2, we generalize notation at Eq. 40 to tensor homomorphisms:

$$\mathcal{W} = \bigoplus_{l=0}^{L-1} \text{Hom}\left(\bigotimes_{v_l} \mathcal{V}_{v_l}^{\text{in}}, \bigotimes_{m_l} \mathcal{V}_{m_l}^{\text{out}}\right) \cong \bigoplus_{l=0}^{L-1} \left[\left(\bigotimes_{m_l} \mathcal{V}_{m_l}^{\text{out}}\right) \otimes \left(\bigotimes_{v_l} \left(\mathcal{V}_{v_l}^{\text{in}}\right)^*\right)\right]. \tag{41}$$

The loss function $\mathcal{L}(\boldsymbol{w})$ (and seems like network itself) is invariant under group actions which in turn act on the hidden layer vector spaces $\{\mathcal{V}_l\}$. Let $\text{Aut}(\mathcal{V}_l)$ denote the group of automorphisms at layer $l$, i.e. transformations that preserve the structure of the layer with respect to the loss. Note that when a basis of single layer $\mathcal{V}_l$ vector space changes, it changes together with the rows of the incoming weights $W_{l-1}$ and the columns of the outgoing weights $W_l$. For a standard feedforward architecture, layer-wise tranformations of corresponding vector spaces are independent, giving to the *global* symmetry group $\mathcal{G}$ the structure of a *direct product*:

$$\mathcal{G} = \prod_{l=0}^{L} \text{Aut}(\mathcal{V}_l). \tag{42}$$

The topology of other architectures, e.g., ResNets, will reinforce constraints, i.e. $\mathcal{V}_l \equiv \mathcal{V}_{l+k}$ for some $l$ and $k \leq L - l$. That changes the simple direct product structure of the global group $\mathcal{G}$ to a *fiber product* or diagonal subgroup where the automorphisms for coupled layers must coincide.

We denote an action of $\mathcal{G}$ on the parameters $\boldsymbol{w} \in \mathcal{W}$ by the representation $\rho : \mathcal{G} \to \text{GL}(\boldsymbol{w})$. It is easy to establish following our previous definitions of direct product and sum, that as $\mathcal{W}$ is a direct sum, $\rho$ decomposes into a *direct sum* of representations. Also, we constrain group representations to be orthogonal such that $\rho_l(a_l)^{-1} = \rho_l(a_l)^\top$. Weight tensors order-2, i.e. matrices $W_l$, transform via conjugation by the group elements of its input and output spaces, and therefore the global representation of $\mathcal{G}$:

$$\rho(a) = \bigoplus_{l=0}^{L-1} (\rho_{l+1}(a_{l+1}) \otimes \rho_l(a_l)). \tag{43}$$

Here, the term $(\rho_l(a_{l+1}) \otimes \rho_l(a_l))$ captures the simultaneous basis change on the domain and codomain of $W_l$. Note that we used the definition of conjugate representations as $\rho^*(a) = \rho(a^{-1})$ and orthogonality of representations $\rho$ to cancel transposes (inverses) at Eq. 43.

**Structure of global network commutant**   Now, let us analyze the influence of the representation $\rho$ on the structure of the commutant matrix $S$ such that $\rho(a)S = S\rho(a)$ for all $a \in \mathcal{G}$. As established in Eq. 43, the global representation decomposes into a direct sum. The matrix $S$ inherits this structure.

We derive the block structure of $S$ by vectorizing the invariance condition $S = \rho(a)S\rho(a)^\top$, which transforms into a linear fixed-point equation acting on the vectorized matrix:

$$\rho(a)^{\otimes 2}\text{vec}(S) = \text{vec}(S), \quad \forall a \in \mathcal{G}. \tag{44}$$

Now, we applying the distributivity of the tensor product over the direct sum and therefore expanding the operator $\rho(a)^{\otimes 2}$:

$$\rho^{\otimes 2} = \left(\bigoplus_l \rho_{l+1} \otimes \rho_l\right)^{\otimes 2} = \bigoplus_{l,k} \underbrace{(\rho_{l+1} \otimes \rho_l) \otimes (\rho_{k+1} \otimes \rho_k)}_{\text{Action on block } S_{lk}}. \tag{45}$$

Equation (45) implies that the global operator $\rho(a)^{\otimes 2}$ block-diagonalizes into $L^2$ independent tensor product actions, each corresponding to a pair of layers with indices $l$ and $k$. Consequently, the high-dimensional fixed-point equation decouples:

instead of solving for the full matrix $S$ simultaneously, we can solve independent commutant equations for each block $S_{lk}$ corresponding to the interaction between layer $l$ and layer $k$.

Eq. 45 can be generalized to any weight tensors:

$$\rho^{\otimes 2} = \bigoplus_{l,k} \left[ \left( \bigotimes_{v_l} \rho_{v_l} \right) \otimes \left( \bigotimes_{m_k} \rho_{m_k} \right) \right]. \tag{46}$$

### D.4. Schur-Weyl Duality and Diagram Solutions

**Theorem D.5** (Schur-Weyl duality by Weyl (1946)). *Let $V = \mathbb{R}^d$ be a vector space and $V^{\otimes k}$ be the k-fold tensor product space. We consider two representations on this space: coordinate and permutation representations. Let any $g \in \mathrm{GL}_d$, let $\rho(a)$ be the matrix representing the simultaneous transformation of all tensor factors, then coordinate representation is defined as the k-fold Kronecker product $\rho(a)^k = \otimes_{i=1}^{k} \rho(a)$.*

*Now let any $\sigma \in S_k$, such that $\pi(\sigma)$ is the permutation matrix that reorders the tensor indices. Then permutation representation is defined as $\pi(\sigma)$ which is the unique linear map defined by its action on the tensor basis elements $e_{i_1} \otimes \cdots \otimes e_{i_k}$:*

$$\pi(\sigma)(e_{i_1} \otimes e_{i_2} \otimes \cdots \otimes e_{i_k}) = e_{i_{\sigma^{-1}(1)}} \otimes \cdots \otimes e_{i_{\sigma^{-1}(k)}}. \tag{47}$$

*Therefore, a linear operator $T$ acting on $V^{\otimes k}$ is invariant under all coordinate transformations, in other words it commutes with $\rho(a)$ for all $a \in \mathrm{GL}_d$, if and only if it is a linear combination of permutation matrices.*

$$T\rho(a)^k = \rho(a)^k T \quad \Longleftrightarrow \quad T = \sum_{\sigma \in S_k} \alpha_\sigma \pi(\sigma) \tag{48}$$

*where $\alpha_\sigma \in \mathbb{R}$ are scalar coefficients.*

Theorem D.5 is fundamental concept that explains how to bridge two different types of symmetry: continuous and permutation symmetry when both act on the same space. In fact, the true power of Schur-Weyl duality is found in ability to transform continuous high dimensional problems into a small combinatorial one.

Classical Schur-Weyl duality between $\mathrm{GL}_n$ and $S_k$ has been extended with the support of many other symmetries. For example, duality has been established for orthogonal groups with Brauer algebra (Doty and Hu, 2009), for permutation groups, including its subgroups, with corresponding partition algebras (Halverson and Ram, 2005). An interesting fact that by restricting the symmetry group we strictly enlarge the algebra size:

$$\mathrm{GL}_n \supseteq O_n \supseteq \quad B_n \supseteq \quad S_n \tag{49}$$
$$\mathcal{S}_k \subseteq \mathcal{B}_k \subseteq \mathcal{A}_{k+\frac{1}{2}} \subseteq \mathcal{A}_k, \tag{50}$$

where $\mathrm{GL}_n$, $O_n$, $B_n$ and $S_n$ are general linear, orthogonal, signed permutation, and permutation groups respectively; $\mathcal{S}_k$, $\mathcal{B}_k$, $\mathcal{A}_{k+\frac{1}{2}}$ and $\mathcal{A}_k$ are groups' corresponding algebras. In other words, the algebra becomes more complex and the number of basis increases. Moreover, we know exactly how many bases of solutions each algebra has. Table 1 provides sizes of Brauer and partition algebras for matching orthogonal and permutation groups.

**Diagrammatic approach** As we have seen, our problem is solving global fixed-point equation (Eq. 44 that in turn decomposes into smaller independent fixed-point equations using Eq. 46. Instead of solving those equations element-by-element using direct numerical algorithms, these fixed-point equations can be solved by constructing the combinatorial basis

*Table 1.* Symmetry Groups and Commutant Algebras. The column $D_k$ lists the number of independent basis solutions to the fixed-point equation $\rho(a)^{\otimes 2k}\mathrm{vec}(T) = \mathrm{vec}(T)$. Here, $B_{2k}$ denotes the Bell number.

| Activation | Symmetry Group $\mathcal{G}$ | Commutant Algebra | Basis Dimension ($D_k$) |
|---|---|---|---|
| linear | $O_n$ Orthogonal Group | $\mathcal{B}_k(n)$ Brauer Algebra | $(2k-1)!!$ |
| relu | $S_n$ Permutation Group | $\mathcal{A}_k(n)$ Partition Algebra | $B_{2k}$ |

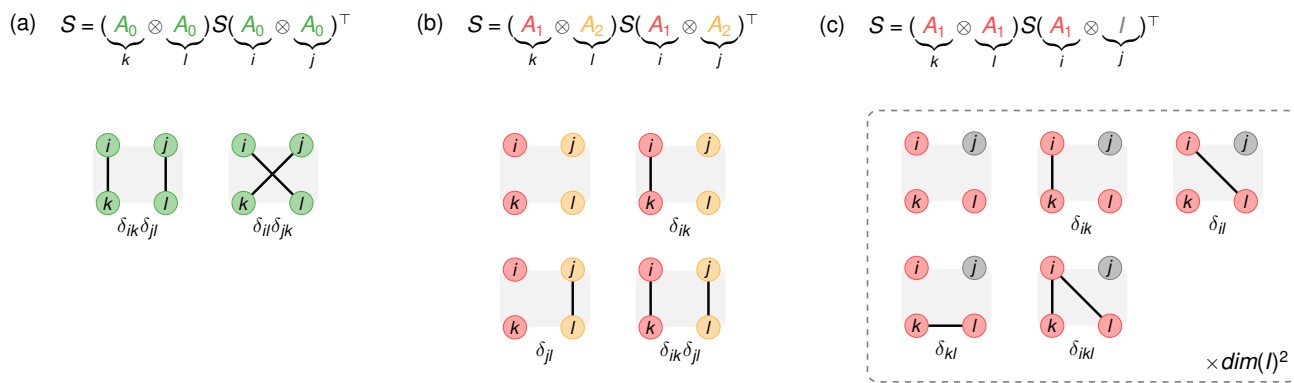

*Figure 7.* Examples of diagrammatic bases for commutant algebra under different symmetry constraints. **(a)** Solution space for $\mathrm{GL}_d$ fixed-point equation $A_0^{\otimes 4}\mathrm{vec}(S) = \mathrm{vec}(S)$, where $A_0 \equiv \rho(a)$ for $a \in \mathrm{GL}_d$, the solution space is spanned by $S_2$. **(b)** Mixed symmetry example for fixed-point equation $(A_1 \otimes A_2)^{\otimes 2}\mathrm{vec}(S) = \mathrm{vec}(S)$, where $A_1 \equiv \rho(a)$ for $a \in S_d$, $A_2 \equiv \rho(a)$ for $a \in S_{d'}$, where $S_d$ and $S_{d'}$ are permutation groups. The indices of $S$ are split into two independent sets, and nodes in this diagram are connected by principle like-to-like forbidding mixed connections. **(c)** Partial symmetry (identity map) example is a special case of (b). The diagram for fixed-point equation $(A_1 \otimes I \otimes A_1^{\otimes 2})\mathrm{vec}(S) = \mathrm{vec}(S)$ is split into two sets: *active* and *passive*. The passive index $j$ allows arbitrary connectivity and hence it tensors the diagrammatic basis of active set with $\dim(I) \times \dim(I)$ matrix.

of the corresponding algebras, per Theorem D.5. We employ a combinatorial approach that utilizes visual schemes called diagrams (Bowman, 2025), which were inspired by Penrose graphical notation for tensors.

Let us build our intuition for how these diagrams work by applying them to a specific form of fixed-point equation $\rho(a)^{\otimes 2k}\mathrm{vec}(T) = \mathrm{vec}(T)$. In this framework, the diagram consists of two rows with $k$ nodes representing column and row indices of the tensor. A basis element of the algebra is constructed simply by linking these column and row nodes. The specific algebras define the rules for how those links allowed to be established. Let us show which rules are used in the case different algebras:

- $\mathrm{GL}_n$. In case of GL symmetry, we are allowed to link nodes in 1 to 1 fashion. Every node in a row can connect only to a single node from another row, and no horizontal connections are allowed.

- $O_n$. Diagrams for orthogonal group with corresponding Brauer algebra follow the same rules for GL symmetry with an exception that we can connect horizontal nodes.

- $S_n$. In case of permutation group, the restriction for linking nodes vanish almost entirely. This diagram corresponds to set partitions. We can group nodes however we like: merge some nodes into a single cluster, leave a node disconnected, connect all nodes.

Fig. 7(a) depicts diagrams that correspond to bases spanning commutant algebra of $\rho(a)^{\otimes 4}\mathrm{vec}(S) = \mathrm{vec}(S)$. $S$ is an operator of size $d^2$ by $d^2$ that commutes with the $\mathrm{GL}_d$ group action. Diagrams show an application of Theorem D.5 which tells us that the solution for $\mathrm{vec}(S)$ is spanned by permutation group $S_2$ which acts on indices of tensor $S$, i.e. its rows and columns. The connections in diagram are interpreted as corresponding products of $\delta$ functions: $\delta_{ik}\delta_{jl}$ and $\delta_{il}\delta_{jk}$. Those $\delta$ combinations have matrix forms: $\delta_{ik}\delta_{jl}$ is an identity map (diagonal matrix) and $\delta_{il}\delta_{jk}$ is a swap operator (commutation matrix). Therefore $S$ is a linear combination of these two:

$$S = \alpha_1 I + \alpha_2 K, \tag{51}$$

where $\alpha_1 \in \mathbb{R}$ and $\alpha_2 \in \mathbb{R}$ are coefficients.

**Diagrams for local network commutants**  So far we have considered uniform Kronecker products, such as presented in our simple example $\rho(a)^{\otimes 4}\mathrm{vec}(S) = \mathrm{vec}(S)$, Fig. 7(a). However, the structure of each local fixed point equation at Eq. 46 can have much reacher structure. This structure is defined by invariant transformations applied to the network weights. We describe a diagrammatic approach here for such local fixed point equations. Our focus is interaction between two layers with order-2 weights, e.g. gradient covariance between $n^{\text{th}}$ and $m^{\text{th}}$ layer, which we denote in the main text as $S^{(nm)}$, and

for brevity we will use $S$ instead for the rest of this section. The fixed point equation with diverse structure looks as:

$$\left(\bigotimes_{i=1}^{4} \rho_i(a_i)\right) \text{vec}(S) = \text{vec}(S).$$
(52)

We distinguish three sets of configurations: **1)** representations are all the same, which we considered in the first place; **2)** the representation is a direct product of subgroups, i.e. mixed representations; and **3)** the presence of identity maps (partial symmetry), where one or more $\rho_i(a_i) \equiv I$.

The first case has been considered before, see Fig. 7(a), i.e. the single group determines the algebra (e.g., orthogonal group with related Brauer algebra), and therefore we follow classical combinatorial approach for finding bases of commutant space. Next, we specifically address the diagrammatic rules for cases 2 and 3.

- **Mixed representations** (Goodman et al., 2009). Consider the fixed-point equation generated by a product of two independent groups, $(A_1 \otimes A_2)^{\otimes 2} \text{vec}(S) = \text{vec}(S)$, see Fig. 7(b), where $A_1 \equiv \rho(a_1)$ for $a_1 \in S_d$, and $A_2 \equiv \rho(a_2)$ for $a_2 \in S_{d'}$. Here, the tensor indices can be thought of as having different "colors" or types corresponding to their respective groups. For instance, at Fig. 7 permutation group $S_d$ acts on $i$ and $k$ indices, and $S_{d'}$ acts on $j$ and $l$ indices. Group colors govern the connectivity of nodes in the diagram, and it is forbidden to link nodes with different colors. Sets of common group are treated as independent systems, and nodes are connected following the combinatorial pattern of corresponding commutant algebra.

- **Partial symmetry**. Identity maps splits the diagram into two sets: *passive* and *active*. In case of the passive set, i.e. the identity map, symmetry constraints vanish, therefore there is total freedom to connect nodes. In contrast, nodes members of the active set follow the connectivity rules of corresponding groups (commutant algebra). The identity map has a multiplicative effect on the solution space. Fig. 7(c) depicts an diagram example for fixed-point equation $(A_1 \otimes I \otimes A_1^{\otimes 2})\text{vec}(S) = \text{vec}(S)$, where $A_1 \equiv \rho(a)$, and $a \in S_d$. Here we take the standard diagram the active permutation group with 3 nodes at indices $i$, $k$, $l$, and tensor it with any matrix $M$ on index $j$. The reason for introducing weight matrix $M$ that commutation of the form $MI = IM$ holds for any matrix $M$, so the solution for identity map is entire $\dim(I) \times \dim(I)$ space. This is equivalent to copying active part of the diagram and weight each copy with entries of the $M$ matrix.

There rules generalize to any structured fixed-point equation $\bigotimes_{i=1}^{k} \rho_i(a_i)\text{vec}(S) = \text{vec}(S)$.

### D.5. Structure of the commutant algebra

In this section, we detail our approach for generalizing the examples of Sec. 3.2 and elucidating the structure of orbit averages in the general case, based on the insights presented in Appendix D.4. Specifically, for two (vectorized) tensors $\boldsymbol{v}$ and $\boldsymbol{v}'$ subject to a transformation group $\mathcal{G}$ as introduced in Sec. 2.1, $S_{\boldsymbol{v}\boldsymbol{v}'} \equiv \mathcal{R}_2(\boldsymbol{v}, \boldsymbol{v}', \mathcal{G})$ satisfies the equation

$$S_{\boldsymbol{v}\boldsymbol{v}'} = \left(\bigotimes_{k=1}^{d} A_{i(k)}\right) S_{\boldsymbol{v}\boldsymbol{v}'} \left(\bigotimes_{k=1}^{d'} A_{i'(k)}\right)^{\top}.$$
(53)

for any $\{A_1, A_2, \ldots\}$ in the group. Note that first-order orbit averages are also covered by this formalism, by considering the case where $\boldsymbol{v}'$ is a 'null' tensor of order $d' = 0$. Solutions to Eq. 12 are order-$(d + d')$ tensors that belong to the group's commutant (or centralizer) algebra, and whose structured can be derived through a diagrammatic approach (Fig. 8). For each unique $A_j$, we begin by identifying all axes of $\boldsymbol{v}$ and $\boldsymbol{v}'$ which the group action transforms through the same $A_j$. In other words, we identify the left- and right- axis sets $\alpha(j) \equiv \{k; i(k) = j\}$ and $\beta(j) \equiv \{k; i'(k) = j\}$.

The sets $\alpha(0)$ and $\beta(0)$ play a special role: it is clear that Eq. 12 does not constrain the solution $S$ along any of its axes that belongs to $\alpha(0) \cup \beta(0)$ (recall that $A_0 = I$ by convention). Thus, any solution must include a sub-tensor of to-be-determined factors $f_{\alpha(0) \cup \beta(0)}$. The homogeneous nature of Eq. 12 means that, even if $\alpha(0) \cup \beta(0) = \varnothing$, all solutions still involve a scalar factor $f$.

In contrast, solutions are very strongly constrained along the other axes. The approach we detail here applies to the case where all transformations $A_j$ are permutations, but larger groups are easy to treat too and yield further restrictions to the

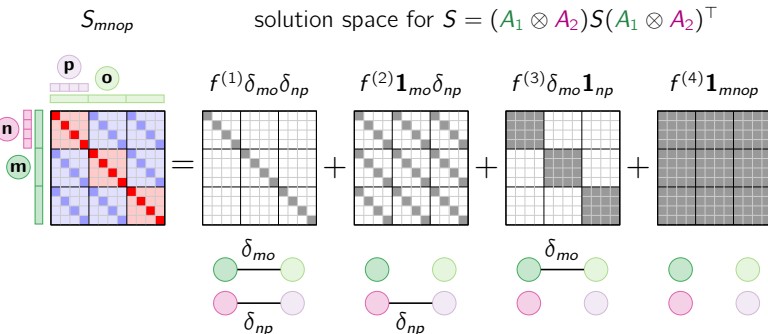

*Figure 8.* Example showing the structure of the commutant algebra for $(\mathcal{G}_1 \times \mathcal{G}_2)$ where $\mathcal{G}_1$ and $\mathcal{G}_2$ are the symmetric groups in dimension 3 and 4, respectively. Any tensor $S$ that satisfies $S = (A_1 \otimes A_2)S(A_1 \otimes A_2)$ for any $(A_1, A_2) \in \mathcal{G}$ is a linear combination of four basis tensors, represented diagrammatically through specific partitions of the tensor axes. See Appendix D.5 for details.

space of solutions. For a given $j$ and associated axis sets $\alpha(j)$ and $\beta(j)$, solutions to Eq. 12 can take any of $B_{|\alpha(j) \cup \beta(j)|}$ forms[2] along the axes concerned. These solution components are given by all possible ways of partitioning $\alpha(j) \cup \beta(j)$. Each partition groups specific axes together, and each group contributes to the solution through a Kronecker delta function over the corresponding indices. Singleton axes in any partition do not contribute (i.e. contribute a $1$ – c.f. $\mathbf{1}_{np}$ in the example above). The space of solutions is then given by a cartesian product of all possible ways of choosing a partition for each of the $A_j$'s. This diagrammatic approach is illustrated in Fig. 8 for an example where $i(1) = i'(1) = 1$ and $i(2) = i'(2) = 2$.

The following table shows the structure of the commutant algebras associated with all possible ways of transforming a pair of parameter tensors (line-separated sections of the table). Here, we restrict the parameter tensors to order $\leq 2$, i.e. vectors or matrices (higher-order cases can be treated using the exact same approach but would yield too large a table). $A_1$ and $A_2$ denote two independent permutations, and $I$ denotes the identity matrix of the appropriate dimension. The way to interpret e.g. the third row is as follows: if, for any possible choice of a permutation matrix $A_1 \in \mathbb{R}^{n \times n}$, we have $S = A_1 S A_1^\top$, then the matrix $S \in \mathbb{R}^{n \times n} = \{S_{ik}\}$ must be a linear superposition of two terms: a scalar factor $c^{(1)}$ times the matrix full of ones, and another scalar factor $c^{(2)}$ times the identity matrix ($\delta_{ik}$).

| left tensor | right tensor | commutation equation | solution form | solution basis |
|---|---|---|---|---|
| vector | vector | $S = ISI^\top$ | $S_{ik}$ | $c^{(1)}_{ik}$ |
| vector | vector | $S = ISA_1^\top$ | $S_{ik}$ | $c^{(1)}_i$ |
| vector | vector | $S = A_1 S A_1^\top$ | $S_{ik}$ | $c^{(1)}$ |
| | | | | $c^{(2)}\delta_{ik}$ |
| vector | vector | $S = A_1 S A_2^\top$ | $S_{ik}$ | $c^{(1)}$ |
| vector | matrix | $S = IS(I \otimes I)^\top$ | $S_{ik\ell}$ | $c^{(1)}_{ik\ell}$ |
| vector | matrix | $S = IS(I \otimes A_1)^\top$ | $S_{ik\ell}$ | $c^{(1)}_{ik}$ |
| vector | matrix | $S = IS(A_1 \otimes I)^\top$ | $S_{ik\ell}$ | $c^{(1)}_{i\ell}$ |
| vector | matrix | $S = IS(A_1 \otimes A_1)^\top$ | $S_{ik\ell}$ | $c^{(1)}_i$ |
| | | | | $c^{(2)}_i \delta_{k\ell}$ |
| vector | matrix | $S = IS(A_1 \otimes A_2)^\top$ | $S_{ik\ell}$ | $c^{(1)}_i$ |
| vector | matrix | $S = A_1 S(I \otimes I)^\top$ | $S_{ik\ell}$ | $c^{(1)}_{k\ell}$ |
| vector | matrix | $S = A_1 S(I \otimes A_1)^\top$ | $S_{ik\ell}$ | $c^{(1)}_k$ |
| | | | | $c^{(2)}_k \delta_{i\ell}$ |
| vector | matrix | $S = A_1 S(I \otimes A_2)^\top$ | $S_{ik\ell}$ | $c^{(1)}_k$ |
| vector | matrix | $S = A_1 S(A_1 \otimes I)^\top$ | $S_{ik\ell}$ | $c^{(1)}_\ell$ |

---

[2] $B_k$ denotes the $k^{\text{th}}$ Bell number.

|  |  |  |  | $c_\ell^{(2)}\delta_{ik}$ |
| --- | --- | --- | --- | --- |
| vector | matrix | $S = A_1 S (A_1 \otimes A_1)^\top$ | $S_{ik\ell}$ | $c^{(1)}$ |
|  |  |  |  | $c^{(2)}\delta_{i\ell}$ |
|  |  |  |  | $c^{(3)}\delta_{k\ell}$ |
|  |  |  |  | $c^{(4)}\delta_{ik}$ |
|  |  |  |  | $c^{(5)}\delta_{ik\ell}$ |
| vector | matrix | $S = A_1 S (A_1 \otimes A_2)^\top$ | $S_{ik\ell}$ | $c^{(1)}$ |
|  |  |  |  | $c^{(2)}\delta_{ik}$ |
| vector | matrix | $S = A_1 S (A_2 \otimes I)^\top$ | $S_{ik\ell}$ | $c_\ell^{(1)}$ |
| vector | matrix | $S = A_1 S (A_2 \otimes A_2)^\top$ | $S_{ik\ell}$ | $c^{(1)}$ |
|  |  |  |  | $c^{(2)}\delta_{k\ell}$ |
| vector | matrix | $S = A_2 S (I \otimes A_1)^\top$ | $S_{ik\ell}$ | $c_k^{(1)}$ |
| vector | matrix | $S = A_2 S (A_1 \otimes I)^\top$ | $S_{ik\ell}$ | $c_\ell^{(1)}$ |
| vector | matrix | $S = A_2 S (A_1 \otimes A_1)^\top$ | $S_{ik\ell}$ | $c^{(1)}$ |
|  |  |  |  | $c^{(2)}\delta_{k\ell}$ |
| vector | matrix | $S = A_2 S (A_1 \otimes A_2)^\top$ | $S_{ik\ell}$ | $c^{(1)}$ |
|  |  |  |  | $c^{(2)}\delta_{i\ell}$ |
| matrix | matrix | $S = (I \otimes I) S (I \otimes I)^\top$ | $S_{ijk\ell}$ | $c_{ijk\ell}^{(1)}$ |
| matrix | matrix | $S = (I \otimes I) S (I \otimes A_1)^\top$ | $S_{ijk\ell}$ | $c_{ijk}^{(1)}$ |
| matrix | matrix | $S = (I \otimes I) S (A_1 \otimes I)^\top$ | $S_{ijk\ell}$ | $c_{ij\ell}^{(1)}$ |
| matrix | matrix | $S = (I \otimes I) S (A_1 \otimes A_1)^\top$ | $S_{ijk\ell}$ | $c_{ij}^{(1)}$ |
|  |  |  |  | $c_{ij}^{(2)}\delta_{k\ell}$ |
| matrix | matrix | $S = (I \otimes I) S (A_1 \otimes A_2)^\top$ | $S_{ijk\ell}$ | $c_{ij}^{(1)}$ |
| matrix | matrix | $S = (I \otimes A_1) S (I \otimes A_1)^\top$ | $S_{ijk\ell}$ | $c_{ik}^{(1)}$ |
|  |  |  |  | $c_{ik}^{(2)}\delta_{j\ell}$ |
| matrix | matrix | $S = (I \otimes A_1) S (I \otimes A_2)^\top$ | $S_{ijk\ell}$ | $c_{ik}^{(1)}$ |
| matrix | matrix | $S = (I \otimes A_1) S (A_1 \otimes I)^\top$ | $S_{ijk\ell}$ | $c_{i\ell}^{(1)}$ |
|  |  |  |  | $c_{i\ell}^{(2)}\delta_{jk}$ |
| matrix | matrix | $S = (I \otimes A_1) S (A_1 \otimes A_1)^\top$ | $S_{ijk\ell}$ | $c_i^{(1)}$ |
|  |  |  |  | $c_i^{(2)}\delta_{j\ell}$ |
|  |  |  |  | $c_i^{(3)}\delta_{k\ell}$ |
|  |  |  |  | $c_i^{(4)}\delta_{jk}$ |
|  |  |  |  | $c_i^{(5)}\delta_{jk\ell}$ |
| matrix | matrix | $S = (I \otimes A_1) S (A_1 \otimes A_2)^\top$ | $S_{ijk\ell}$ | $c_i^{(1)}$ |
|  |  |  |  | $c_i^{(2)}\delta_{jk}$ |
| matrix | matrix | $S = (I \otimes A_1) S (A_2 \otimes I)^\top$ | $S_{ijk\ell}$ | $c_{i\ell}^{(1)}$ |
| matrix | matrix | $S = (I \otimes A_1) S (A_2 \otimes A_2)^\top$ | $S_{ijk\ell}$ | $c_i^{(1)}$ |
|  |  |  |  | $c_i^{(2)}\delta_{k\ell}$ |
| matrix | matrix | $S = (I \otimes A_2) S (A_1 \otimes A_1)^\top$ | $S_{ijk\ell}$ | $c_i^{(1)}$ |
|  |  |  |  | $c_i^{(2)}\delta_{k\ell}$ |
| matrix | matrix | $S = (I \otimes A_2) S (A_1 \otimes A_2)^\top$ | $S_{ijk\ell}$ | $c_i^{(1)}$ |

| | | | | |
|---|---|---|---|---|
| | | | | $c_i^{(2)}\delta_{j\ell}$ |
| matrix | matrix | $S = (A_1 \otimes I)S(I \otimes A_2)^\top$ | $S_{ijk\ell}$ | $c_{jk}^{(1)}$ |
| matrix | matrix | $S = (A_1 \otimes I)S(A_1 \otimes I)^\top$ | $S_{ijk\ell}$ | $c_{j\ell}^{(1)}$ |
| | | | | $c_{j\ell}^{(2)}\delta_{ik}$ |
| matrix | matrix | $S = (A_1 \otimes I)S(A_1 \otimes A_1)^\top$ | $S_{ijk\ell}$ | $c_j^{(1)}$ |
| | | | | $c_j^{(2)}\delta_{i\ell}$ |
| | | | | $c_j^{(3)}\delta_{k\ell}$ |
| | | | | $c_j^{(4)}\delta_{ik}$ |
| | | | | $c_j^{(5)}\delta_{ik\ell}$ |
| matrix | matrix | $S = (A_1 \otimes I)S(A_1 \otimes A_2)^\top$ | $S_{ijk\ell}$ | $c_j^{(1)}$ |
| | | | | $c_j^{(2)}\delta_{ik}$ |
| matrix | matrix | $S = (A_1 \otimes I)S(A_2 \otimes I)^\top$ | $S_{ijk\ell}$ | $c_{j\ell}^{(1)}$ |
| matrix | matrix | $S = (A_1 \otimes I)S(A_2 \otimes A_2)^\top$ | $S_{ijk\ell}$ | $c_j^{(1)}$ |
| | | | | $c_j^{(2)}\delta_{k\ell}$ |
| matrix | matrix | $S = (A_1 \otimes A_1)S(A_1 \otimes A_1)^\top$ | $S_{ijk\ell}$ | $c^{(1)}$ |
| | | | | $c^{(2)}\delta_{i\ell}$ |
| | | | | $c^{(3)}\delta_{j\ell}$ |
| | | | | $c^{(4)}\delta_{k\ell}$ |
| | | | | $c^{(5)}\delta_{ik}$ |
| | | | | $c^{(6)}\delta_{ik\ell}$ |
| | | | | $c^{(7)}\delta_{ik}\delta_{j\ell}$ |
| | | | | $c^{(8)}\delta_{jk}$ |
| | | | | $c^{(9)}\delta_{i\ell}\delta_{jk}$ |
| | | | | $c^{(10)}\delta_{jk\ell}$ |
| | | | | $c^{(11)}\delta_{ij}$ |
| | | | | $c^{(12)}\delta_{ij\ell}$ |
| | | | | $c^{(13)}\delta_{ij}\delta_{k\ell}$ |
| | | | | $c^{(14)}\delta_{ijk}$ |
| | | | | $c^{(15)}\delta_{ijk\ell}$ |
| matrix | matrix | $S = (A_1 \otimes A_1)S(A_1 \otimes A_2)^\top$ | $S_{ijk\ell}$ | $c^{(1)}$ |
| | | | | $c^{(2)}\delta_{ik}$ |
| | | | | $c^{(3)}\delta_{jk}$ |
| | | | | $c^{(4)}\delta_{ij}$ |
| | | | | $c^{(5)}\delta_{ijk}$ |
| matrix | matrix | $S = (A_1 \otimes A_1)S(A_2 \otimes A_2)^\top$ | $S_{ijk\ell}$ | $c^{(1)}$ |
| | | | | $c^{(2)}\delta_{ij}$ |
| | | | | $c^{(3)}\delta_{k\ell}$ |
| | | | | $c^{(4)}\delta_{ij}\delta_{k\ell}$ |
| matrix | matrix | $S = (A_1 \otimes A_2)S(A_1 \otimes A_2)^\top$ | $S_{ijk\ell}$ | $c^{(1)}$ |
| | | | | $c^{(2)}\delta_{ik}$ |
| | | | | $c^{(3)}\delta_{j\ell}$ |

| | | | | $c^{(4)}\delta_{ik}\delta_{j\ell}$ |
| --- | --- | --- | --- | --- |
| matrix | matrix | $S = (A_1 \otimes A_2)S(A_2 \otimes A_2)^\top$ | $S_{ijk\ell}$ | $c^{(1)}$ |
| | | | | $c^{(2)}\delta_{j\ell}$ |
| | | | | $c^{(3)}\delta_{k\ell}$ |
| | | | | $c^{(4)}\delta_{jk}$ |
| | | | | $c^{(5)}\delta_{jk\ell}$ |
| matrix | matrix | $S = (A_2 \otimes I)S(A_1 \otimes A_1)^\top$ | $S_{ijk\ell}$ | $c_j^{(1)}$ |
| | | | | $c_j^{(2)}\delta_{k\ell}$ |
| matrix | matrix | $S = (A_2 \otimes I)S(A_1 \otimes A_2)^\top$ | $S_{ijk\ell}$ | $c_j^{(1)}$ |
| | | | | $c_j^{(2)}\delta_{i\ell}$ |

## E. Proof of Lemma 3.2

We rewrite the definition of orbit average

$$\mathcal{R}_1(\boldsymbol{v}, \mathcal{G}) \equiv \mathbb{E}_\mathcal{G}\left[\left(\bigotimes_{k=1}^d A_{i(k)}\right)\boldsymbol{v}\right] = \mathbb{E}_\mathcal{G}(A\boldsymbol{v}) \quad \text{with} \quad A \equiv \bigotimes_{k=1}^d A_{i(k)}. \tag{54}$$

This is a linear operator acting on the vectorized tensor $\boldsymbol{v}$, that can be represented by the matrix $P$:

$$\mathcal{R}_1(\boldsymbol{v}, \mathcal{G}) \equiv P\boldsymbol{v} \quad \text{with} \quad P \equiv \frac{1}{|\mathcal{G}|}\sum_j A^{(j)}. \tag{55}$$

The index $j$ runs over all members of the group $\mathcal{G}$, and we recall that $\mathcal{G} = \mathcal{G}_0 \times \mathcal{G}_1 \times \ldots \mathcal{G}_m$ is the direct product of groups acting on all tensors, and is itself a group.

We prove that $P$ is an orthogonal projection, namely that $P^2 = P = P^\top$. We start by computing $P^2$. We note that, for any group, given indices $i$ and $j$, there exists an index $k$ such that $A^{(i)}A^{(j)} = A^{(k)}$ and each element of the group occurs $|\mathcal{G}|$ times when summing over both indices $i$ and $j$. Thus,

$$P^2 = \frac{1}{|\mathcal{G}|^2}\sum_{ij} A^{(i)}A^{(j)} = \frac{1}{|\mathcal{G}|}\sum_k A^{(k)} = P. \tag{56}$$

Therefore, the matrix $P$ is a projection. Next, we prove that the matrix $P$ is symmetric or, equivalently, that the projection is orthogonal. By assumption, $\mathcal{G}$ is a subset of the orthogonal group, therefore $A^\top = A^{-1}$ for the matrix representation of any member of $\mathcal{G}$. Furthermore, each element in a group has a unique and distinct inverse within the group. Therefore,

$$P^\top = \frac{1}{|\mathcal{G}|}\sum_i \left(A^{(i)}\right)^\top = \frac{1}{|\mathcal{G}|}\sum_i \left(A^{(i)}\right)^{-1} = \frac{1}{|\mathcal{G}|}\sum_k A^{(k)} = P. \tag{57}$$

Since $P$ is an orthogonal projection, the group average of any tensor is equal to the nearest invariant in square norm.

## F. Rank deficiency of $S_{\boldsymbol{w}}$

In this section, we carry out a simple test for whether $H_g^\star$ achieving better approximation to the Hessian compared to $H_{\text{PD}}^\star$ in Fig. 4 is due to the rank deficiency of $S_{\boldsymbol{w}}$ in Eq. 10. For the same four-layer MLP student-teacher setup as used in Hessian similarity experiments, we compute the singular values spectra of $S_{\boldsymbol{w}}$ across training iterations for three different groups: $[P; P; P], [P; I; P], [I; P; I]$, with $I$ the identity group and $A$ the permutation group. As shown in Fig. 9, $S_{\boldsymbol{w}}$ is indeed heavily rank-deficient in our setup, and becomes increasingly so for smaller groups (note that singular values were computed in single precision, so singular values past 8 orders of magnitude of decay are practically zero; note also that the far tail of the singular value spectrum had to be excluded from our logarithmic visualizations). Therefore, new evidence confirms that inversion of $S_w$ amplifies noise along singular directions which makes $H_{\text{PD}}^\star$ less stable than $H_g^\star$.

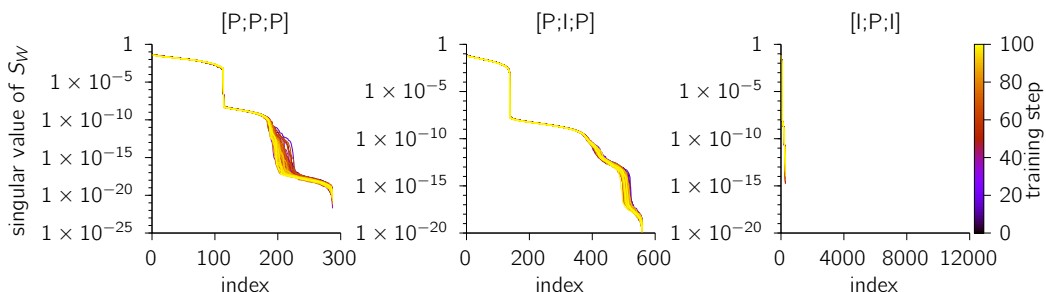

*Figure 9.* Rank deficiency of surrogate matrices.

# G. The Symo optimizer

In Sec. 3.4 we defined the vanilla Symo update as

$$\boldsymbol{w}_{t+1} = \boldsymbol{w}_t - \eta \big(H_{\boldsymbol{g}}^{\star}\big)^{-1}\boldsymbol{g}_t, \tag{58}$$

where $H_{\boldsymbol{g}}^{\star}$ is defined in Eq. 11 and $\boldsymbol{g}_t = \nabla\mathcal{L}(\boldsymbol{w}_t)$ denotes the gradient at iteration $t$. In practice, training heuristics such as damping and momentum can be applied to accelerate convergence and stabilize training.

## G.1. Damping

Computing the update requires inverting $H_{\boldsymbol{g}}^{\star}$ (Eq. 15), which may be ill-conditioned. Damping can be applied via

$$H_{\boldsymbol{g}}^{\star} \leftarrow H_{\boldsymbol{g}}^{\star} + \lambda s_{\max} I, \tag{59}$$

before inversion, where $\lambda$ is the damping parameter and $s_{\max}$ is the largest singular value of $H_{\boldsymbol{g}}^{\star}$.

## G.2. Momentum and bias correction for factors

To improve factor estimation and reduce noise, particularly in stochastic settings, we can apply momentum to the factor estimates, following (Bernacchia, 2025; Appendix K). For a factor $F$, the momentum update is given by

$$\hat{F}_{t+1} = \mu_{\text{factor}}\hat{F}_t + (1 - \mu_{\text{factor}})F_t, \tag{60}$$

where $F_t$ is the factor estimated using the gradient at iteration $t$ (i.e., $\boldsymbol{g}_t$), $\hat{F}_t$ denotes the running average at iteration $t$, and $\mu_{\text{factor}}$ is the momentum parameter for factors. All factors are initialized to zero at $t = 0$, and the estimates are bias corrected according to

$$\hat{F}_{t+1} \leftarrow \frac{\hat{F}_{t+1}}{1 - \mu_{\text{factor}}^t}, \tag{61}$$

as in standard practice (Kingma, 2014).

## G.3. Momentum and bias correction for gradient

In addition to applying momentum to factor estimates, we can also apply momentum and bias correction directly to the gradients. This is done via

$$\hat{\boldsymbol{g}}_{t+1} = \mu_{\boldsymbol{g}}\hat{\boldsymbol{g}}_t + (1 - \mu_{\boldsymbol{g}})\boldsymbol{g}_t, \tag{62}$$

where $\boldsymbol{g}_t$ is the gradient at iteration $t$, $\hat{\boldsymbol{g}}_t$ is its running average, and $\mu_{\boldsymbol{g}}$ is the gradient momentum parameter. We initialize $\hat{\boldsymbol{g}}_0 = 0$, and apply bias correction as

$$\hat{\boldsymbol{g}}_{t+1} \leftarrow \frac{\hat{\boldsymbol{g}}_{t+1}}{1 - \mu_{\boldsymbol{g}}^t}, \tag{63}$$

following standard practice (Kingma, 2014).

# H. Symo-muon-shampoo equivalence

In Sec. 3.4, we showed that the Hessian approximation $H_g^\star = S_g^{\frac{1}{2}}$ is equivalent to the Muon/Shampoo update under specific choices of network architecture and symmetry groups. In this section we provide the detailed derivations for two concrete examples: a multi-layer perceptron (MLP) and a transformer layer.

## H.1. Shampoo and Muon equivalence

In Sec. 2.3 we proved that Shampoo (Gupta et al., 2018) and Muon (Jordan et al., 2024) produce indeed equivalent parameter updates. Here we provide more details for the proof.

$$W_{t+1} = W_t - \eta L_t^{-\frac{1}{4}} G_t R_t^{-\frac{1}{4}}, \qquad \text{Shampoo} \qquad (64)$$
$$W_{t+1} = W_t - \eta \operatorname{pol}(M_t), \qquad \text{Muon} \qquad (65)$$

where $W \in \mathbb{R}^{n \times m}$ and $\eta$ is the learning rate. By default, Muon utilizes a momentum buffer $M_t = \beta M_{t-1} + (1 - \beta)G_{t-1}$ and the unitary polar factor $\operatorname{pol}(M) = UV^\top$ derived from the singular value decomposition $M = U\Sigma V^\top$ (Amsel et al., 2025). On the other hand, Shampoo uses cumulative tensors $L_{t+1} = L_t + G_t G_t^\top$, $R_{t+1} = R_t + G_t^\top G_t$, which deviates from the Muon setup. For simplicity, we assume that no momentum treatment is applied and focus on $M_t = G_t$ for Muon and $L_t = G_t G_t^\top$, $R_t = G_t^\top G_t$ for Shampoo.

The goal is to demonstrate that the Shampoo update is an exact expansion of the polar factor of the gradient. First, we perform a singular value decomposition on the gradient matrix $G_t = U\Sigma V^\top$. We exploit identity for matrix analytic functions $f$ from Higham (2008, Corollary 1.34), which states:

$$f(AB)A = Af(BA). \qquad (66)$$

Substituting (66) with $A = G$ and $B = G^\top$ into the Shampoo direction term in (64):

$$\begin{aligned}
L_t^{-\frac{1}{4}} G_t R_t^{-\frac{1}{4}} &= (G_t G_t^\top)^{-\frac{1}{4}} G_t (G_t^\top G_t)^{-\frac{1}{4}} \\
&= G_t (G_t^\top G_t)^{-\frac{1}{4}} (G_t^\top G_t)^{-\frac{1}{4}} \\
&= G_t (G_t^\top G_t)^{-\frac{1}{2}}.
\end{aligned} \qquad (67)$$

Recognizing that $(G_t^\top G_t)^{-\frac{1}{2}} = (V\Sigma^2 V^\top)^{-\frac{1}{2}} = V\Sigma^{-1}V^\top$, we substitute the SVD of $G_t$ into (67):

$$\begin{aligned}
G_t (G_t^\top G_t)^{-\frac{1}{2}} &= (U\Sigma V^\top)(V\Sigma^{-1}V^\top) \\
&= U(\Sigma\Sigma^{-1})V^\top \\
&= UV^\top = \operatorname{pol}(G_t).
\end{aligned} \qquad (68)$$

Thus, the Shampoo optimizer update with no accumulation of gradients is algebraically equivalent to the Muon optimizer in the zero momentum limit. We summarize their equivalency of updates via:

$$L_t^{-\frac{1}{4}} G_t R_t^{-\frac{1}{4}} = G_t (G_t^\top G_t)^{-\frac{1}{2}} = \operatorname{pol}(G_t). \qquad (69)$$

## H.2. Feed-forward networks

In Sec. 2.3 we proved that Symo and Shampoo/Muon updates are equivalent under certain constraints. Here, we provide a concrete example using a two-layer bias-free MLP with `tanh` activations. Let the layer widths be $d^{(1)}$ and $d^{(2)}$, and $d^{(0)}$ denotes the input dimension. The largest symmetry group associated with the input and output layers is the identity group, and for the hidden layers it is the signed permutation group (because `tanh` is an odd function applied elementwise).

Ignoring off-diagonal blocks, the gradient covariance matrix $S_g$ can be written as

$$S_g \approx \begin{pmatrix} S_g^{(11)} & 0 \\ 0 & S_g^{(22)} \end{pmatrix}. \qquad (70)$$

As shown in Sec. 3.2, $S_{\boldsymbol{g}}$ satisfies the commutation condition in Eq. 12. Consider the weight updates associated with the first-layer weight matrix $W^{(1)} \in \mathbb{R}^{d^{(1)} \times d^{(0)}}$. The corresponding condition for $S_{\boldsymbol{g}}^{(11)}$ is

$$S_{\boldsymbol{g}}^{(11)} = (I \otimes B_n) \, S_{\boldsymbol{g}}^{(11)} \, (I \otimes B_n)^{\top}, \tag{71}$$

where $I$ denotes the identity symmetry group and $B_n$ denotes the signed permutation symmetry group. This condition admits a unique solution for all $B_n$, namely

$$S_{\boldsymbol{g}}^{(11)} = F \otimes I, \tag{72}$$

where $F \in \mathbb{R}^{d^{(0)} \times d^{(0)}}$ is the free parameter to be optimized and $I \in \mathbb{R}^{d^{(1)} \times d^{(1)}}$ is the identity matrix. To obtain the factor $F$, we solve the associated least-squares problem in the smaller-dimension surrogate space with the assumption that the orbit average of the gradients is equal to zero ($\boldsymbol{g}^{\star} = \boldsymbol{0}$ in Eq. 9c; Sec. 3.3; Bernacchia, 2025). Eq. 13b has the form:

$$F^{\star} = \arg\min_{F} \left| F \otimes I - \boldsymbol{g}^{(1)} \left( \boldsymbol{g}^{(1)} \right)^{\top} \right|_{F}^{2}, \tag{73}$$

where $\boldsymbol{g}^{(1)}$ is the gradient of the loss w.r.t. the weight vector $\boldsymbol{w}^{(1)}$. This equation admits a unique closed-form solution

$$F = \frac{1}{d^{(1)}} \left( G^{(1)} \right)^{\top} G^{(1)}, \tag{74}$$

where $G^{(1)}$ is the gradient of the loss w.r.t. $W^{(1)}$ and $\mathrm{vec}(G^{(1)}) = \mathrm{vec}(G^{(1)})$. The resulting Symo update is

$$
\begin{aligned}
\mathrm{vec}(\Delta G^{(1)}) &= \left( S_{\boldsymbol{g}}^{(11)} \right)^{-\frac{1}{2}} \boldsymbol{g}^{(1)} \\
&= (F \otimes I)^{-\frac{1}{2}} \boldsymbol{g}^{(1)} \\
&= \sqrt{d^{(1)}} \, \mathrm{vec} \left( G^{(1)} \left( \left( G^{(1)} \right)^{\top} G^{(1)} \right)^{-\frac{1}{2}} \right),
\end{aligned} \tag{75}
$$

Thus

$$\Delta G^{(1)} = \sqrt{d^{(1)}} G^{(1)} \left( \left( G^{(1)} \right)^{\top} G^{(1)} \right)^{-\frac{1}{2}}. \tag{76}$$

This gradient update is equivalent to the middle expression at Eq. 69, which proves our statement that the Shampoo / Muon update direction recovers the Symo update, up to a constant scaling factor when momentum is disabled. The proof of update equivalence for $\boldsymbol{w}^{(2)}$ follows analogously, with the final Symo update being

$$\Delta G^{(2)} = \sqrt{d^{(1)}} \left( G^{(2)} \left( G^{(2)} \right)^{\top} \right)^{-\frac{1}{2}} G^{(2)}. \tag{77}$$

This example also illustrates why `tanh` activations, and hence the signed permutation symmetry group for the hidden layer, are necessary. Any other choice of commutation conditions admits different solutions, leading to updates in different forms.

## H.3. Transformer

Here we show that our symmetry aware optimization framework (Symo) matches Shampoo/Muon updates on Transformer model (Vaswani et al., 2017). We consider one layer of the Transformer:

$$X_1 = W_e X + W_{\text{pe}} P_{\text{pe}} \tag{78}$$

$$K = W_k X_1, \quad Q = W_q X_1, \quad V = W_v X_1 \tag{79}$$

$$X_2 = V \left( \frac{\text{softplus}(QK^\top)}{\sqrt{d}} \right) \tag{80}$$

$$X_3 = W_p X_2 \tag{81}$$

$$X_4 = \text{LayerNorm}(X_3 + X_1) \tag{82}$$

$$X_5 = \tanh(W_{\text{ff}_1} X_4) \tag{83}$$

$$X_6 = W_{\text{ff}_2} X_5 \tag{84}$$

$$X_7 = \text{LayerNorm}(X_6 + X_1) \tag{85}$$

$$Y = W_o X_7, \tag{86}$$

where $W_e$, $W_p$ and $W_o$ are the embedding, attention projection and output projection weights respectively and $W_{\text{ff}_1}$ and $W_{\text{ff}_2}$ are feedforward weights and $d$ is the embedding dimension. Following Muon's optimization strategy neither embedding weights, biases, layer normalization parameters nor projection weights are tuned with Muon (they are instead tuned with Adam). This puts an identity map constraint on the choice of the group transformation applied either to weight's columns and rows. When this constraint is applied, the Transformer weights after permutation become:

$$\begin{aligned} W_k' &= A_k W_k I \\ W_q' &= A_k W_q I \\ W_v' &= A_v W_v I \\ W_p' &= I W_p A_v \\ W_{\text{ff}_1}' &= A_{\text{ff}_1} W_{\text{ff}_1} I \\ W_{\text{ff}_2}' &= I W_{\text{ff}_2} A_{\text{ff}_1}, \end{aligned} \tag{87}$$

where $A_k, A_v, A_{\text{ff}_1}$ are permutation matrices. Given the vectorized weight vector $\boldsymbol{w} = [\text{vec}(W_k); \text{vec}(W_q); \ldots, \text{vec}(W_{\text{ff}_2})]$ and the corresponding transformed weight vector $\boldsymbol{w}' = [\text{vec}(W_k'); \text{vec}(W_q'); \ldots, \text{vec}(W_{\text{ff}_2}')]$, the transformations in Eq. 87 can be expressed as a single linear map $\boldsymbol{w}' = A\boldsymbol{w}$ where

$$A \equiv \begin{pmatrix} I \otimes A_k & 0 & 0 & 0 & 0 & 0 \\ 0 & I \otimes A_q & 0 & 0 & 0 & 0 \\ 0 & 0 & I \otimes A_v & 0 & 0 & 0 \\ 0 & 0 & 0 & A_v \otimes I & 0 & 0 \\ 0 & 0 & 0 & 0 & I \otimes A_{\text{ff}_1} & 0 \\ 0 & 0 & 0 & 0 & 0 & A_{\text{ff}_1} \otimes I \end{pmatrix}. \tag{88}$$

We now derive invariant gradient covariance $S = \mathcal{R}(\boldsymbol{w}, \boldsymbol{w}, \mathcal{G})$, that satisfies commutation equation $S = ASA^\top$ (Sec. 3.2).

To obtain equivalence with Shampoo/Muon, we again restrict $S$ to be *block-diagonal*. This simplification reduces the matrix equation $S = ASA^\top$ to a set of independent equations for each block:

$$S^{(ii)} = A^{(ii)} S^{(ii)} (A^{(ii)})^\top,$$

where $A^{(ii)}$ denotes the $i$-th block of the block-diagonal matrix $A$, and $S^{(ii)}$ is the corresponding block of $S$. Each block $A^{(ii)}$ takes the form

$$A^{(ii)} = I \otimes X \quad \text{or} \quad A^{(ii)} = X \otimes I,$$

with $X \in \{A_k, A_v, A_{\text{ff}_1}\}$. Here, $A_k$ and $A_v$ are representations of the orthogonal group, whereas $A_{\text{ff}_1}$ is a representation of the signed permutation group. Since the solutions for orthogonal and signed permutation groups share the same form and derivation, we treat them equivalently.

To solve for blocks $S^{(ii)}$, we reuse a result that was established in Appendix H.2:

$$\begin{cases} S^{(ii)} = F \otimes I, & F = \frac{1}{d^{(i)}} \left(G^{(i)}\right)^\top G^{(i)} & \text{when } A^{(ii)} = I \otimes X \\ S^{(ii)} = I \otimes F, & F = \frac{1}{d^{(i-1)}} G^{(i)} \left(G^{(i)}\right)^\top & \text{when } A^{(ii)} = X \otimes I \end{cases}, \tag{89}$$

where $G^{(i)} \in \mathbb{R}^{d^{(i)} \times d^{(i-1)}}$ denotes the gradient of the loss w.r.t. the weight $W^{(i)}$. Substituting these solutions into corresponding Symo updates we obtain:

$$\begin{cases} (S^{(ii)})^{-\frac{1}{2}} \text{vec}(G^{(i)}) = \frac{1}{\sqrt{d^{(i)}}} \text{vec}\left(G^{(i)} \left((G^{(i)})^\top G^{(i)}\right)^{-\frac{1}{2}}\right) & \text{when } S^{(ii)} = F \otimes I \\ (S^{(ii)})^{-\frac{1}{2}} \text{vec}(G^{(i)}) = \frac{1}{\sqrt{d^{(i-1)}}} \text{vec}\left(\left((G^{(i)}(G^{(i)})^\top\right)^{-\frac{1}{2}} G^{(i)}\right) & \text{when } S^{(ii)} = I \otimes F \end{cases}, \tag{90}$$

which is equivalent to the middle expression at Eq. 69, proving our statement that the Shampoo/Muon update direction recovers the Symo update, up to a constant scaling factor (that is different per layer) and in the absence of momentum.

## I. Details of symmetry group used in Fig. 3

In Fig. 3, we showed the $S_{\boldsymbol{g}} = \mathcal{R}_2(\boldsymbol{g}, \boldsymbol{g}, \mathcal{G})$ for a single-layer Transformer. In this case, we considered the following group $\mathcal{G}$:

$$W_{\text{emb}} \to A_d W_{\text{emb}} I_{\text{vocab}} \tag{91}$$
$$W_{QK} \to \tilde{A}_d W_{QK}(I_2 \otimes A_d) \tag{92}$$
$$W_V \to A_v W_V \tilde{A}_d \tag{93}$$
$$W_{\text{proj}} \to A_d W_{\text{proj}} A_v \tag{94}$$
$$W_{\text{ff}_1} \to A_{\text{f}} W_{\text{ff}_1} A_d \tag{95}$$
$$W_{\text{ff}_2} \to A_d W_{\text{ff}_2} A_{\text{f}}, \tag{96}$$
$$\tag{97}$$

where $A_d$, $A_v$ and $A_{\text{f}}$ are appropriately sized permutation matrices, and $\tilde{A}_d$ is an orthogonal matrix. The transformer was run on a toy dataset generated with uniformly sampled inputs and outputs of vocabulary size equal to 7 and sequence length 50. Note that $W_{QK}$ is a matrix with two matrices as block diagonals each representing weights $W_Q$, and $W_K$ of query and key correspondingly. This construction is necessary following the implementation details in Karpathy, 2026.

## J. Details on Hessian approximation experiments

In this section, we describe the details of the Hessian approximation experiments presented in Sec. 4.

### J.1. Network

We train a four-layer multilayer perceptron (MLP) in a teacher-student setup, mapping the input $x \in \mathbb{R}^{100}$ through layer widths $70 \to 70 \to 70 \to 40$ using `tanh` activation functions. The total number of parameters is 19,850.

Both the teacher network weights and the initial student network weights are drawn from a Gaussian distribution with zero mean and standard deviation $1/\sqrt{d}$, where $d$ denotes the input dimension of the corresponding weight matrix. The network is trained to minimize the mean squared error (MSE) between its output and that of the teacher network. Training with Adam is performed in a full-batch setting with batch size 5000 and a learning rate of 0.1.

As discussed in Sec. 3.4, to demonstrate Symo-Muon-Shampoo equivalence we only consider signed permutation symmetries applied to alternating layers. In particular, we use identity symmetry groups for the input and output weights, and assigning a signed permutation group to every subsequent odd-indexed layer and identity symmetry group for even-indexed layer.

### J.2. Similarity test

The comparison is carried out by computing the cosine similarity between

$$\boldsymbol{g}' - \boldsymbol{g} \quad \text{and} \quad \hat{H}(\boldsymbol{w}' - \boldsymbol{w}),$$

where $w$ denotes the vectorization of parameter $W$ at a given point during optimization, $w' = w + \Delta w$, $g = \nabla\mathcal{L}(w)$, and $g' = \nabla\mathcal{L}(w')$. The cosine similarity between two vectors $\mathbf{a}$ and $\mathbf{b}$ is defined as

$$\cos(\mathbf{a}, \mathbf{b}) = \frac{\mathbf{a} \cdot \mathbf{b}}{\|\mathbf{a}\|\|\mathbf{b}\|}$$

where $\cdot$ is the Euclidean dot product. We consider two choices for $\Delta w$. In the first, we take a step of size $r$ in a random direction,

$$\Delta w = r\,\bar{\epsilon}, \qquad \epsilon \sim \mathcal{N}(0, I),$$

and in the second, we take a step of size $r$ in the negative gradient direction,

$$\Delta w = -r\,\bar{g},$$

where $\bar{x} = x/\|x\|$ such that $\bar{x}$ has unit norm. We use $r = 0.01$ and $N = 1000$ in all runs.

### J.3. Constructing Hessian approximations

We compare against the true Hessian at $w$ ($H = \nabla^2\mathcal{L}(w)$) five other Hessian approximations:

1. the true Hessian evaluated at the orbit-averaged parameters, $H^\star$ (Eq. 7);

2. $H_{\mathrm{PD}}^\star$ (Eq. 10);

3. $H_g^\star$ (Eq. 11);

4. block-diagonal $H_g^\star$ (denoted as $H_g^\star$ (BD));

5. the Shampoo Hessian approximation (Sec. 2.3).

**Shampoo and Symo Hessian-vector products**    In Sec. 3.4 and Appendix H, we showed that the block-diagonal Symo update with curvature matrix $H_g^\star(\mathrm{BD})$ is mathematically equivalent to the Shampoo update under certain restrictions. However, this equivalence holds at the level of the *update direction* and does not imply that the corresponding $\hat{H}\,v$ products are identical for arbitrary $v \neq g$.

Consider the block of $\hat{H}\,v$ associated with the the $i$-th layer weight matrix $W \in \mathbb{R}^{d^{(i)} \times d^{(i-1)}}$. Since both the Shampoo curvature matrix and $H_g^\star(\mathrm{BD})$ employ block-diagonal Hessian approximations, each block of $\hat{H}\,v$ can be analyzed independently. For notational simplicity, we henceforth omit the block indices. For Shampoo, the matrix–vector product of a particular block takes the form

$$\hat{H}\,v = v\left(L^{\frac{1}{4}} V R^{\frac{1}{4}}\right)$$
$$= \mathrm{vec}\left((GG^\top)^{\frac{1}{4}} V (G^\top G)^{\frac{1}{4}}\right) \tag{98}$$

where $V \in \mathbb{R}^{d^{(i)} \times d^{(i-1)}}$ is the unvectorized version of $v$. In contrast, the corresponding Symo product is given by

$$H_g^\star(\mathrm{BD})\,v = (S_g)^{\frac{1}{2}}\,v$$
$$= (F \otimes I)^{\frac{1}{2}}\,v$$
$$= \frac{1}{\sqrt{d^{(i)}}}\,\mathrm{vec}\left(V (G^\top G)^{\frac{1}{2}}\right), \tag{99}$$

or

$$H_g^\star(\mathrm{BD})\,v = (S_g)^{\frac{1}{2}}\,v$$
$$= (I \otimes F)^{\frac{1}{2}}\,v$$
$$= \frac{1}{\sqrt{d^{(i-1)}}}\,\mathrm{vec}\left((GG^\top)^{\frac{1}{2}}V\right), \tag{100}$$

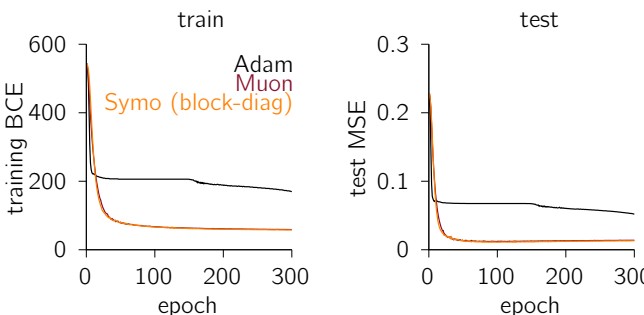

*Figure 10.* We train an MNIST autoencoder with Symo, Muon and Adam. Results for **(Left)** training binary cross-entropy loss (BCE); **(Right)** test mean-squared error loss (MSE).

where we make use of the expressions for $F$ derived in Appendix H. Equation (98) and Equations (100) (99) are equivalent if and only if

$$(GG^\top)^{\frac{1}{4}} V = V (G^\top G)^{\frac{1}{4}}, \tag{101}$$

which holds, for example, when $V$ is a scaled multiple of $G$, but not for a general matrix $V$. In practice, an appropriate scaling factor $\frac{1}{\sqrt{d^{(i)}}}$ or $\frac{1}{\sqrt{d^{(i-1)}}}$ was applied to the corresponding blocks of $H_g^\star(\mathrm{BD})$ to demonstrate that Symo matches Shampoo in the gradient direction (Fig. 4 (right)).

In our hessian approximation experiments, all covariance matrices (i.e., $S_g$ and $S_w$) are damped as $X \leftarrow X + \lambda I$ with $\lambda = 10^{-6}$ prior to any inversion or square-root operations.

## K. Details on equivalence experiments

In this section, we describe the details of the two experiments carried out for demonstrating Symo-Shampoo-Muon equivalence in Sec. 5: the MNIST autoencoder with an MLP and the character-level prediction task with nanoGPT.

### K.1. Details on Autoencoder experiments

In this section, we describe the details of the MNIST autoeconder experiments presented in Sec. 5.1.

#### K.1.1. NETWORK

We study a deep autoencoder trained on the MNIST handwritten digit dataset (LeCun et al., 1998), using the same experimental configuration as in Martens and Grosse (2015). The model is a fully connected autoencoder with a symmetric encoder–decoder topology. All layers are affine, including bias parameters, and employ tanh activation functions, except for the final reconstruction layer, which uses a sigmoid nonlinearity. Inputs $x \in \mathbb{R}^{784}$ are mapped through hidden layer dimensions $784 \to 1000 \to 500 \to 250 \to 30 \to 250 \to 500 \to 1000 \to 784$. The network contains a total of 2,837,314 trainable parameters. Training is performed using full-batch optimization over the 60,000 training images for 300 epochs, and evaluation is conducted on a held-out test set of 10,000 images. The learning objective is the binary cross-entropy loss between the input and its reconstruction.

#### K.1.2. OPTIMIZERS

This section details the specific configurations of optimizers used in MNIST autoencoder experiments. Exact hyperparameter values used and ranges explored can be found in the tables in Appendix K.3.

**Muon** We apply the standard PyTorch implementation of Muon (Jordan et al., 2024; Liu et al., 2025) to all two-dimensional weights in the autoencoder. Since Muon is not defined for one-dimensional parameters, we apply Adam to the bias parameters. To have a fair comparison with Symo we disable weight decay as well as Nesterov momentum in Muon. For this experiment we set the number of Newton-Schulz iteration steps to 10.

**Symo** As discussed in Sec. 3.4, Symo and Muon coincide under a set of well-defined conditions. For the MNIST autoencoder, these conditions include:

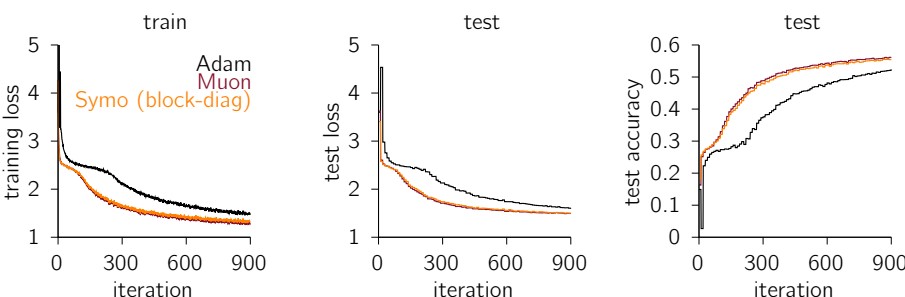

*Figure 11.* We train a nanoGPT on Shakespeare dataset character-level prediction task with Symo, Muon and Adam optimizers. Results for **(Left)** training loss; **(Middle)** test loss; **(Right)** test accuracy.

1. Using a block-diagonal $H_g^\star$ as the curvature matrix.

2. Setting the symmetry on the input space to be trivial by taking the input group as the identity, and imposing a signed permutation symmetry on each subsequent odd-indexed layer.

3. Removing the scaling factor for Symo updates (i.e., $\sqrt{d^{(i)}}$ in Eq. 76).

4. Removing the scaling factor for Muon updates (i.e., $\max\left(1, \sqrt{\text{fan}_{\text{out}}/\text{fan}_{\text{in}}}\right)$ (Jordan et al., 2024)).

5. Replacing the Newton–Schulz method for computing the update parameter (Jordan et al., 2024) with the exact inverse square-root of the outer products of the gradient vector, effectively recovering the Shampoo update (Sec. 2.3).

6. Use Adam for optimizing one-dimensional parameters (i.e. biases) and Symo/Muon for two-dimensional parameters.

7. When calculating $H_g^\star$ we assume the orbit average of the gradients is equal to zero ($g^\star = 0$).

In our experiments, we enforce Item 1, Item 2 and Item 7 but do not enforce the rest in order to compare the theoretically sound Symo optimization step with the empirically effective Muon update. For Item 3, since the scaling factor differs across gradient vectors, exact matching would require different learning rates for each vector. Instead, we use a single global learning rate for Symo, tuned to best approximate the Muon update. For Item 4 and Item 5 we retain the original Muon implementations, which are known to perform well in practice. As for Item 6, we retain the Muon implementation, which uses Adam to optimize one-dimensional parameters, while in the Symo run we still apply Symo updates to these parameters. We also adopt Muon's momentum formulation, applying momentum to the gradients prior to factor estimation. As shown in Fig. 10, despite these implementation differences, we are able to closely match the Muon training and evaluation curves.

**Adam**  We compare results with the standard PyTorch implementation of Adam (Kingma, 2014).

### K.2. Details on nanoGPT experiments

In this section, we describe the details of the nanoGPT experiments presented in Sec. 5.2.

#### K.2.1. NETWORK

We train a small nanoGPT model (Karpathy, 2026) on the Tiny Shakespeare character-level prediction task (Karpathy, 2015). The network consists of 6 transformer blocks, each with 6 attention heads and an embedding dimension of $384$. The total number of learnable parameters of 10.6 million. The vocabulary size is 65 and sequence length is 256. Training is performed in a mini-batch setting with batch size 64, and early stopping with max patience of ten is applied using validation loss as the stopping criterion. Dropout is set to $0.2$ in all experiments.

#### K.2.2. OPTIMIZERS

This section details the specific configurations of optimizers used in nanoGPT experiments. Exact hyperparameter values used and ranges explored can be found in the tables in Appendix K.3.

**Muon**   Following the standard PyTorch implementation of Muon (Jordan et al., 2024; Liu et al., 2025), we apply Muon to parameters with dimension greater than one (e.g., weights in MLP layers and attention projections), and use Adam for one-dimensional parameters (e.g., biases). To demonstrate the equivalence between Muon and Symo, we set the weight decay to $0$ and disable Nesterov momentum. We use the default value of five Newton-Schulz steps.

**Symo**   As outlined in Sec. 3.4, Symo and Muon are equivalent under specific conditions. For the nanoGPT experiment we summarize these conditions here:

1. Using a block-diagonal $H_g^\star$ as the curvature matrix.

2. Apply the trivial identity group for embedding weights, orthogonal group symmetries among the key, query, and value weights, and permutation group symmetries across the attention heads.

3. Removing the scaling factor for Symo updates (i.e., $\sqrt{d^{(1)}}$ in Eq. 76).

4. Removing the scaling factor for Muon updates (i.e., $\max\left(1, \sqrt{\mathrm{fan_{out}}/\mathrm{fan_{in}}}\right)$ (Jordan et al., 2024)).

5. Replacing the Newton–Schulz method for computing the update parameter (Jordan et al., 2024) with the exact inverse square-root of the outer products of the gradient vector, effectively recovering the Shampoo update (Sec. 2.3).

6. Use Adam for optimizing one-dimensional parameters (i.e. embedding weights, biases, layer normalization) and Symo/Muon for two-dimensional parameters.

7. When calculating $H_g^\star$ we assume the orbit average of the gradients is equal to zero ($g^\star = 0$).

In our nanoGPT experiments, we enforce Item 1, Item 6 Item 7 in List K.2.2. However, similar to the autoencoder experiments, we do not enforce Item 3, Item 4 and Item 5, for the same reasons explained in Appendix K.1. We also adopt Muon's momentum formulation, applying momentum to the gradients prior to factor estimation. As shown in Fig. 11, despite these implementation differences, we are able to closely match the Muon training and test loss curves and test accuracy curve.

The parameter groups used by Symo are chosen according to their underlying symmetry properties. The identity group is used for the embedding layer weights, input weights and output weights, as these parameters do not admit non-trivial symmetry transformations (Appendix H.3). The permutation group is applied to head-group parameters to reflect the invariance of the model to permutations of attention heads. Finally, the orthogonal group is used for attention head parameters, capturing rotational invariances within the attention subspace.

**Adam**   We use the PyTorch implementation of Adam (Kingma, 2014).

### K.3. Hyperparameters

Here we provide details on the method for hyperparameter tuning in the autoencoder and nanoGPT experiments described in Appendix K.1 and Appendix K.2. Hyperparameter optimization was conducted using `Optuna` (Akiba et al., 2019). For each experiment and optimizer configuration, we performed 100 optimization trials, where each trial corresponds to a training run with a distinct set of hyperparameters.

For Symo, Muon, and Adam, hyperparameters were sampled from fixed search spaces, with parameter ranges reported in Table 4, Table 6 and Table 8. The optimization objective was to minimize the validation loss evaluated at a fixed number of training steps. For the autoencoder experiments, the validation loss after 30 epochs was used. For the nanoGPT experiments, the validation loss after 500 training iterations was used.

`Optuna`'s default sampler was used, which is based on Bayesian optimization via the Tree-structured Parzen Estimator (TPE). For each experiment, the final hyperparameter configuration was selected as the trial achieving the lowest validation loss across the 100 trials. The final hyperparameters used are reported in Table 3, Table 5 and Table 7 for Symo, Muon and Adam respectively.

*Table 3.* Hyperparameters used for Symo. $\eta_{\text{symo}}$ is the learning rate for Symo (for training parameters with dimension greater than one) and $\eta_{\text{adam}}$ is the learning rate for Adam (For training one-dimensional parameters). $\beta_1$ and $\beta_2$ are coefficients used for computing running averages of gradient and its square respectively for the Adam optimizer. $\mu_{\boldsymbol{g}}$ is the momentum parameter on gradient and $\lambda$ is the damping parameter for Symo. We do not apply momentum on factors (see Appendix G).

| | $\eta_{\text{symo}}$ | $\eta_{\text{adam}}$ | $\beta_1$ | $\beta_2$ | $\mu_{\boldsymbol{g}}$ | $\lambda$ |
|---|---|---|---|---|---|---|
| **Autoencoder** | $4.6 \times 10^{-3}$ | - | - | - | 0.79 | $1.0 \times 10^{-9}$ |
| **NanoGPT** | $6.8 \times 10^{-4}$ | $8 \times 10^{-4}$ | 0.82 | 0.99 | 0.7 | 0 |

*Table 4.* Range of hyperparameters explored for Symo.

| | $\eta_{\text{symo}}$ | $\eta_{\text{adam}}$ | $\beta_1$ | $\beta_2$ | $\mu_{\boldsymbol{g}}$ | $\lambda$ |
|---|---|---|---|---|---|---|
| **Autoencoder** | $1 \times 10^{-4} - 2 \times 10^{-2}$ | - | - | - | $0.7 - 0.99$ | $1 \times 10^{-10} - 1 \times 10^{-8}$ |
| **NanoGPT** | $1 \times 10^{-4} - 2 \times 10^{-2}$ | $5 \times 10^{-4} - 2 \times 10^{-3}$ | $0.8 - 0.9$ | 0.99 | 0.7 | 0 |

*Table 5.* Hyperparameters used for Muon. $\eta_{\text{muon}}$ is the learning rate for Muon (for training parameters with dimension greater than one) and $\eta_{\text{adam}}$ is the learning rate for Adam (For training one-dimensional parameters). $\beta_1$ and $\beta_2$ are coefficients used for computing running averages of gradient and its square respectively for the Adam optimizer. $\mu$ is the momentum for the Muon optimizer.

| | $\eta_{\text{muon}}$ | $\eta_{\text{adam}}$ | $\beta_1$ | $\beta_2$ | $\mu$ | $\epsilon$ |
|---|---|---|---|---|---|---|
| **Autoencoder** | $9.9 \times 10^{-2}$ | $1 \times 10^{-3}$ | 0.9 | 0.999 | 0.79 | $1 \times 10^{-9}$ |
| **NanoGPT** | 0.01 | $8 \times 10^{-4}$ | 0.82 | 0.99 | 0.7 | 0 |

*Table 6.* Range of hyperparameters explored for Muon.

| | $\eta_{\text{muon}}$ | $\eta_{\text{adam}}$ | $\beta_1$ | $\beta_2$ | $\mu$ | $\epsilon$ |
|---|---|---|---|---|---|---|
| **Autoencoder** | $1 \times 10^{-3} - 2 \times 10^{-1}$ | $1 \times 10^{-3}$ | 0.9 | 0.999 | $0.7 - 0.99$ | $1 \times 10^{-10} - 1 \times 10^{-8}$ |
| **NanoGPT** | $1 \times 10^{-3} - 2 \times 10^{-1}$ | $5 \times 10^{-4} - 2 \times 10^{-3}$ | $0.8 - 0.9$ | 0.99 | 0.7 | 0 |

*Table 7.* Hyperparameters used for Adam. $\eta$ is the learning rate. $\beta_1$ and $\beta_2$ are coefficients used for computing running averages of gradient and its square respectively. $\epsilon$ is term added to the denominator in Adam to improve numerical stability.

| | $\eta$ | $\beta_1$ | $\beta_2$ | $\epsilon$ |
|---|---|---|---|---|
| **Autoencoder** | $9 \times 10^{-4}$ | 0.9 | 0.93 | $1 \times 10^{-8}$ |
| **NanoGPT** | $8 \times 10^{-4}$ | 0.82 | 0.99 | $1 \times 10^{-10}$ |

*Table 8.* Range of hyperparameters explored for Adam.

| | $\eta$ | $\beta_1$ | $\beta_2$ | $\epsilon$ |
|---|---|---|---|---|
| **Autoencoder** | $5 \times 10^{-4} - 2 \times 10^{-3}$ | $0.8 - 0.99$ | $0.9 - 0.999$ | $1 \times 10^{-10} - 1 \times 10^{-8}$ |
| **NanoGPT** | $5 \times 10^{-4} - 2 \times 10^{-3}$ | $0.8 - 0.99$ | $0.9 - 0.999$ | $1 \times 10^{-10} - 1 \times 10^{-8}$ |

## L. Details of Sec. 6 – exploration of different group choices

For Fig. 6, we trained MLPs as autoencoders of the MNIST and FACES datasets, a well-established benchmark for second-order optimization (Goldfarb et al., 2020). Network architectures, training losses, and dataset splits were exactly as in Goldfarb et al. (2020). The network architecture, layer sizes and activation functions at each layer are presented in Table 9. The hyperparameters for Symo, Adam and Shampoo are presented in Table 10, Table 11 and Table 12 respectively (only learning rates were optimized by grid search).

*Table 9.* Autoencoder architectures used for MNIST and FACES datasets. Layer sizes are listed from input to bottleneck and back to output.

|  | Input | L1 | L2 | L3 | Bottleneck | L5 | L6 | L7 | Output |
|---|---|---|---|---|---|---|---|---|---|
| **MNIST** | | | | | | | | | |
| Layer sizes | 784 | 1000 | 500 | 250 | 30 | 250 | 500 | 1000 | 784 |
| Activation | | ReLU | ReLU | ReLU | Linear | ReLU | ReLU | ReLU | Sigmoid |
| **FACES** | | | | | | | | | |
| Layer sizes | 625 | 2000 | 1000 | 500 | 30 | 500 | 1000 | 2000 | 625 |
| Activation | | ReLU | ReLU | ReLU | Linear | ReLU | ReLU | ReLU | Linear |

*Table 10.* Hyperparameters used for Symo. $\eta$ is the learning rate. $\mu_{\boldsymbol{g}}$ is the momentum parameter on gradient and $\mu_{\text{factor}}$ is the momentum parameter on factors. $\lambda$ is the damping parameter.

|  | $\eta\,\{\text{MNIST}|\text{FACES}\}$ | $\mu_{\boldsymbol{g}}$ | $\mu_{\text{factor}}$ | $\lambda$ |
|---|---|---|---|---|
| **Symo** | $\{4|2\} \times 10^{-3}$ | 0.9 | 0.9 | $1.0 \times 10^{-4}$ |
| **Symo (larger)** | $\{2|1\} \times 10^{-3}$ | 0.9 | 0.9 | $1.0 \times 10^{-4}$ |
| **Symo (smaller, BD)** | $\{4|2\} \times 10^{-3}$ | 0.9 | 0.9 | $1.0 \times 10^{-4}$ |

*Table 11.* Hyperparameters used for Adam. $\eta$ is the learning rate. $\beta_1$ and $\beta_2$ are coefficients used for computing running averages of gradient and its square respectively. $\epsilon$ is term added to the denominator in Adam to improve numerical stability.

|  | $\eta$ | $\beta_1$ | $\beta_2$ | $\epsilon$ |
|---|---|---|---|---|
| **MNIST** | $4 \times 10^{-4}$ | 0.9 | 0.9 | $1 \times 10^{-4}$ |
| **FACES** | $4 \times 10^{-4}$ | 0.9 | 0.9 | $1 \times 10^{-4}$ |

*Table 12.* Hyperparameters used for Shampoo. $\eta$ is the learning rate. $\beta$ is the momentum parameter. $\epsilon$ is the damping parameter.

|  | $\eta$ | $\beta$ | $\epsilon$ |
|---|---|---|---|
| **MNIST** | $2 \times 10^{-1}$ | 0.9 | $1 \times 10^{-4}$ |
| **FACES** | $2 \times 10^{-1}$ | 0.9 | $1 \times 10^{-4}$ |

## M. Example Code

Here we present an example code snippet for training a three-layer MLP with Symo.

*Listing 1.* Example code for training a simple MLP using the Symo optimizer

```
import torch
```

```python
import torch.nn as nn
from symo.factory import groups_spec
from symo.optim import Symo

class MLP(nn.Module):
    ...
    def forward(self, x):
        x = nn.Linear(input_dim, hidden_dim)(x)
        x = torch.relu(x)
        x = nn.Linear(hidden_dim, hidden_dim)(x)
        x = torch.relu(x)
        x = nn.Linear(hidden_dim, output_dim)(x)
        return x

model = MLP(input_dim, hidden_dim, output_dim)

# [ Group ]_[ ID ]
# Group can be "I" for identity group
# "S" for permutation group
# "B" for signed-permutation group
# "O" for orthogonal group
# ID indicates whether two groups share the same identity
In = "I_input"
Ou = "I_output"
Gh1 = "S_hidden1"
Gh2 = "S_hidden2"

groups = {
    "MLP.0.weight": (Gh1, In), #(out_features, in_features)
    "MLP.0.bias": (Gh1,),
    "MLP.1.weight": (Gh2, Gh1),
    "MLP.1.bias": (Gh2,),
    "MLP.2.weight": (Ou, Gh1),
    "MLP.2.bias": (Ou,),
}
groups_list = [
    groups[name] for n, _ in model.named_parameters()
]

# Each ID should have a dimension size
sizes = {"input": input_dim,
        "output": output_dim,
        "hidden1": hidden_dim,
        "hidden2": hidden_dim,}

optimizer = Symo(
    params=model.parameters(),
    groups_spec=groups_spec(groups_list, sizes),
    lr=learning_rate,
    grads_beta=0.9,
    factors_beta=0.9,
)

...

for _ in range(iterations):
    x,y = next(data_batch)

    optimizer.zero_grad()
    loss = mse_loss(model(x), y)
    loss.backward()
    optimizer.step()
```

