# OpenReview forum: "Exploiting weight-space symmetries for approximating curvature"
_ICML.cc/2026/Conference — ICML 2026 regular_

### Official Review · Reviewer_gmTt · 2026-03-10

**Soundness:** 3
**Presentation:** 3
**Significance:** 2
**Originality:** 3
**Overall Recommendation:** 5
**Confidence:** 4

**Summary:**

The paper studies a structured Hessian approximation through parameter symmetries and introduces an optimizer, called *symo*, based on this. The central idea is to form a symmetry-constrained curvature surrogate by taking a second-order Taylor expansion around the orbit-averaged parameter and then using gradient equivariance to express the resulting surrogate in terms of second-order orbit averages of the weights and gradients. In particular, this Hessian approximation can be represented in a structured low-complexity basis determined by the chosen symmetry group. The authors empirically justify this approximation in the direction of the negative gradient. For a specific choice of symmetry group (signed permutation symmetries acting on alternating layers of an MLP), the resulting approximation recovers Shampoo/Muon up to constant scaling as a special case. Empirically, the method is evaluated on an MNIST autoencoder and NanoGPT, and the results suggest that the symmetry choice underlying Shampoo/Muon provides a favorable tradeoff between approximation quality and computational cost. The accompanying codebase contains a library that  automatically derives the corresponding basis of a given user-specified symmetry group employed by the *symo* Hessian approximation.

**Compliance With Llm Reviewing Policy:**

Affirmed.

**Final Justification:**

See rebuttal acknowledgement.

**Key Questions For Authors:**

- Could you please refer to the points in “Weaknesses”?
- In figure 4, $H_g^\ast$ seems to be a better approximation to the Hessian in the direction of the negative gradient compared to $H_{PD}^\ast$, despite being an approximation of the latter. Section 3.1 hypothesizes that $S_w$ which would have to be inverted to compute $H_{PD}^\ast$, may be rank deficient. Did you investigate this in practice, estimate $S_w$ in some cases, or have other intuitions about why the surrogate outperforms the quantity it is meant to approximate here?
- Lines 340-343 claim that “[cosine] similarities of 0.1-0.2 are quite high considering the high dimensionality of the parameter space”. I think this could be substantiated a bit more. While the baseline is zero, it is not immediately clear to me that this score would mean anything in practice or is just weak positive alignment.

**Limitations:**

yes

**Strengths And Weaknesses:**

Strengths:

- In my opinion, this is an interesting and timely contribution that bridges subfields (parameter symmetry / optimization).
- The representation of group averages via Schur-Weyl duality is quite elegant and, to the best of my knowledge, new in this context.
- This paper also goes significantly beyond the work [1] it builds on, as it systemizes the basis construction and has far more comprehensive evaluation (esp NanoGPT).
- The content is well presented and relatively easy to follow.
- I only skimmed the code, but it looks nicely written and the framework seems quite usable and general, especially considering that this is a more theoretical work.

Weaknesses:

- The authors acknowledge this, but there is no guarantee for whether the Taylor expansion in eq. (7) holds. The authors argue that “smaller orbits will have smaller errors”, but many ML architectures have very large orbits, so $w$ and $w^\ast$ should be vastly different (think, in a 1 hidden layer ReLU MLP, $w^\ast$ would effectively collapse to a single neuron). Furthermore, if I understand correctly, section 3.1 models $H^\ast$ as the Hessian at $w^\ast$ not $w$. If the symmetry group is sufficiently large, it would seem unlikely (or at least, quite surprising) to me that in such cases the Hessian at $w^\ast$ would give any meaningful information about the Hessian at $w$.
- If it is not optimal for the method to choose the largest known parameter symmetry group, then in practice there might be a lot of trial-and-error needed to configure the method for a new architecture.
- The connection to Shampoo/Muon is quite conditional and seems a bit post-hoc: supposing that each block is Kronecker factorized (eq. (16)) and that we consider, e.g., for an MLP, only signed permutation symmetries acting on alternating layers of an MLP (and the identity otherwise), reads like a somewhat arbitrary choice to me. Furthermore, if i understand correctly, e.g. in an MLP, the aforementioned group is not even a parameter symmetry if the nonlinearity is not chosen to be sign equivariant (as e.g. tanh in line 206), but I would imagine the choice on nonlinearity shouldnt have a significant impact on the performance on Muon vs SGD/Adam/etc?

[1] Alberto Bernacchia. *Global curvature for second-order optimization of neural networks.* ICML 2025.

---

> ### Author Rebuttal · Authors · 2026-03-30
>
> We thank Reviewer gmTt for the technical review and the kind remarks about both the Schur–Weyl duality application and the code quality. We address raised concerns below.
>
> **Q (Why does $H^\star_g$ outperform $H^\star_{\mathrm{PD}}$ in Figure 4?).** This is an excellent observation, which we have indeed been able to connect to the rank deficiency of $S_w$ which we have examined in new experiments (see [Fig. 1](https://github.com/anonymous-reviewer-icml-2026/symo/blob/main/s-w-svals.pdf)). For the same four-layer MLP student-teacher setup as used in Hessian similarity experiments (Section 4 and Figure 4 in the main text), we compute the singular value spectra of $S_w$ across training iterations for three different groups ([P;P;P], [P;I;P] and [I;P;I], with I the identity group and P the permutation group). Our experiments show that $S_w$ is indeed heavily rank-deficient in our setup, and becomes increasingly so for smaller groups (note that singular values were computed in single precision, so singular values past 8 orders of magnitude of decay are practically zero; note also that the far tail of the singular value spectrum had to be excluded from our logarithmic visualizations). Therefore, new evidence confirms that inversion of $S_w$ amplifies noise along singular directions which makes $H^\star_{\mathrm{PD}}$ less stable than $H^\star_g$.
>
> **Q (Cosine similarities of $[0.1, 0.2]$ - are these meaningful?).** We agree this claim needs calibration. As a baseline, the expected cosine similarity between two random unit vectors in $\mathbb{R}^d$ concentrates around 0 with standard deviation $1/\sqrt{d}$. For our parameter space $d = 19850$, the standard deviation is approximately $0.007$. Thus, cosine similarities in the range $[0.1, 0.2]$ are well above what is expected from random vectors.
>
> **Q (Nonlinearity dependence, tanh vs ReLU?).** Please see our discussion of W1 in [the reply to Reviewer GZc3](https://openreview.net/forum?id=yiq2tZojBK&noteId=pvSst2IYu0). Importantly, the equivalence appears relatively insensitive to the specific choice of activation: we showed that Muon and Symo behave near-identically on the MNIST autoencoder with ReLU activation (Figure 6), and in nanoGPT with GELU activation, neither of which lead to exact signed permutation invariance. This suggests that the *structure* of the curvature approximation (Kronecker form, Eq. 16) matters more than the formal symmetry justification.
>
> **Q (Trial-and-error for configuring symmetry groups on new architectures?).** This is a valid practical concern. However, 1) users need only specify the group settings, and our PyTorch library automates the derivation of orbit averages; 2) Figure 6 demonstrates that performance degrades gracefully across different group choices, showing that even "wrong" group choices outperform the Adam optimizer. Yet, we admit that group selection for new architectures remains an open question. We will add this to the section on limitations.
>
> **Q (Taylor expansion around the orbit average).** Please see our discussion of W1 in the [reply to Reviewer 8698](https://openreview.net/forum?id=yiq2tZojBK&noteId=FdSxd5rY6r), where we present new empirical results on approximation quality vs. orbit size and a preliminary theoretical bound. The bound confirms the intuition that larger groups lead to larger orbits and therefore less accurate Taylor approximations, while also showing that the Shampoo/Muon group choice occupies a middle ground.

---

> > ### Author Rebuttal · Reviewer_gmTt · 2026-04-03
> >
> > I thank the authors for their thorough rebuttal, and my main points have been adequately addressed. Expanding the current discussion about the rank deficiency of $S_w$ with the new evidence further strengthens the approach taken by the work, so in my view this will be a valuable addition. While I feel that my broader point about the connection to Shampoo/Muon feeling a bit "post-hoc" still stands, the work has clear merit and this should not stand in the way of acceptance. Therefore, I would support acceptance and will raise my score accordingly.

---

### Official Review · Reviewer_4Goq · 2026-03-12

**Soundness:** 3
**Presentation:** 4
**Significance:** 2
**Originality:** 3
**Overall Recommendation:** 4
**Confidence:** 3

**Summary:**

This paper exploits the parameter symmetry to compute the hessian of the model, and proposed the connection of the theory to muon and shampoo

**Compliance With Llm Reviewing Policy:**

Affirmed.

**Final Justification:**

Thanks for the rebuttal. I will keep my score of weak acceptance.

**Key Questions For Authors:**

See weakness

**Limitations:**

Well done

**Strengths And Weaknesses:**

Strength: parameter symmetry is a novel, important, but understudied field, and in my opinion, should be promoted a lot more. The results are interesting and novel to the extent of my knowledge. Figures are well-made, and the paper is well-written

Weakness: two fixable problems: (1) I think this reference should be mentioned and discussed in detail: arxiv.org/abs/2309.16932, esp., look at its discussion on block diagonal Hessians. (2) I think the experiments should be improved, multiple runs and shaded regions should be included in the results

That being said, I think the result is, while interesting, not that significant, and its importance to the field of AI is yet to be demonstrated. For example, the experiments are done only on simple datasets (such as MNIST) and simple models, and trained for a very short period of time (a few hundred). These are insufficient to establish the use and importance of the theory and algorithm. But I do not think these alone are the reasons for rejection. I therefore recommend weak acceptance.

---

> ### Author Rebuttal · Authors · 2026-03-30
>
> We thank Reviewer 4Goq for the positive feedback and for highlighting the importance of parameter symmetry as an understudied field. We are glad the results were found interesting and novel, and we appreciate the kind words about the figures. Below, we address the raised concerns.
>
> **Q (Missing citation).** Thank you for pointing this out. We will add the reference and discuss its relevance to block-diagonal Hessian structure in the revised manuscript.
>
> **Q (Multiple runs with shaded regions. Scale of experiments).** Please see our discussion of W2 in the [reply to Reviewer 8698](https://openreview.net/forum?id=yiq2tZojBK&noteId=FdSxd5rY6r). We will include means $\pm$ 95% confidence intervals over 5 random seeds for all training curves (see [Fig. 1](https://github.com/anonymous-reviewer-icml-2026/symo/blob/main/mnist-nanogpt.pdf), [Fig. 2](https://github.com/anonymous-reviewer-icml-2026/symo/blob/main/similarity-std.pdf)). We respectfully note that the primary contribution is theoretical, and the experiments are designed to validate the theory rather than demonstrate state-of-the-art optimization performance. We will make this positioning clearer in the revised manuscript.

---

> > ### Author Rebuttal · Reviewer_4Goq · 2026-03-31
> >
> > Thanks for the rebuttal. I will keep my score of weak acceptance.

---

### Official Review · Reviewer_GZc3 · 2026-03-13

**Soundness:** 3
**Presentation:** 3
**Significance:** 3
**Originality:** 3
**Overall Recommendation:** 4
**Confidence:** 3

**Summary:**

The paper proposes Symo, a symmetry-aware curvature approximation built by averaging gradient outer products over weight-space symmetry orbits. The main point is that these orbit averages lie in the commutant algebra of the chosen symmetry group, so the resulting curvature proxy is highly structured and can be stored and inverted through a small set of factors. The paper further argues that the chosen symmetry group controls a complexity-accuracy trade-off, and that particular restricted group choices recover Shampoo/Muon-like update directions. Empirically, it reports a toy Hessian-approximation study, training-curve comparisons on an MNIST autoencoder and nanoGPT, and ablations over different symmetry groups.

**Compliance With Llm Reviewing Policy:**

Affirmed.

**Final Justification:**

This paper gives an excellent presentation of an interesting and original idea. The rebuttal addressed my main concerns. I maintain my opinion that this paper deserves acceptance.

**Key Questions For Authors:**

- Would it be possible to generalize the proofs of lemmas 3.1 and 3.2 to continuous groups?
- The Shampoo/Muon connection in Section 3.3 and experiments seems to rely on the assumption that $g^* = 0$. Is this true?

**Limitations:**

yes

**Strengths And Weaknesses:**

### Strengths
- The paper brings an interesting algebraic idea, that symmetry orbit-averaging impose strong structure on second-order objects.
- The topic is important. A principled symmetry-based route to tractable curvature approximations could matter for optimization, uncertainty, pruning, and continual learning, and the connection to Shampoo/Muon is potentially valuable.
- The paper is conceptually original. Using weight-space symmetries to derive approximate curvature, and interpreting Shampoo/Muon as particular symmetry choices, is a novel and interesting viewpoint.
- The paper is clearly written and well organized.


### Weaknesses
- The proofs of the orbit-average lemmas (3.1 and 3.2) are written for finite groups, but later sections use continuous orthogonal groups without pointing this out.
- In eq. (7), there is no explicit bound on how far $w$ can be from $w^*$. As a result, it is not clear how good this approximation is. Since this is the key step in deriving the approximated curvature, it would be helpful to justify the validity of this approximation.
- “Orbit averaging gives rise to Shampoo/Muon” seems overly strong, as symo recovers a special version of Shampoo/Muon that does not exactly match the one used in practice.

---

> ### Author Rebuttal · Authors · 2026-03-30
>
> We thank Reviewer GZc3 for the technical feedback. We appreciate the recognition of the algebraic idea and the potential broader impact of the framework. Below, we address raised concerns.
>
> **W2 (The Shampoo/Muon connection).** We accept the reviewer's criticism and we will soften the claim in the revised manuscript. We also believe that our claims in the main paper are fair, as we carefully list all the assumptions needed to establish the equivalence between Symo and Muon: 1) block-diagonal restriction of the curvature matrix (but not Kronecker factorization, which is a result, not an assumption), 2) sign permutation symmetries on alternating layers (for MLP only; not required for nanoGPT), 3) sign-equivariant activations -- tanh (MLP and nanoGPT). Empirically, the equivalence appears relatively insensitive to the specific choice of nonlinear activation; for example, we showed that Muon and Symo behave near-identically on the MNIST autoencoder network with ReLU activation, and in nanoGPT with GELU activation, neither of which lead to exact signed permutation invariance. As far as the theory goes, whilst we were able to derive Symo/Muon equivalence for two widespread network configurations (MLP and Transformer blocks), it is hard to show that there are no other configurations where the equivalence holds too.
>
> **Q (Generalize Lemmas 3.1 and 3.2 to continuous groups?).** Thank you for correctly noting that Lemmas 3.1 and 3.2 are proved only for finite groups but occasionally applied to continuous orthogonal groups. We will update the proofs in the appendix: both lemmas extend directly to compact groups by replacing the finite sum $\frac{1}{|G|}\sum(\cdot)$ with the Haar integral $\int_G(\cdot)\,\mathrm{d}\mu(g)$. Our two core lemmas, namely that orbit averaging is an orthogonal projection and that orbit averages commute with the group action, follow from the same algebraic argument since compact groups admit a unique normalized bi-invariant Haar measure, and the proof structure (Appendix A, Equation 19) requires only the group closure property, which holds identically under the Haar integral. We will expand our proofs to cover continuous orthogonal groups.
>
> **Q (Bound on the approximation error in Equation 7?).** Please see our discussion of W1 in [the reply to Reviewer 8698](https://openreview.net/forum?id=yiq2tZojBK&noteId=FdSxd5rY6r), where we present both new empirical measurements of approximation quality vs. orbit size and a preliminary theoretical bound based on the dimension of the $G$-invariant subspace.
>
> **Q (Does the Shampoo/Muon connection rely on $g^\star = 0$?).** Yes.

---

> > ### Author Rebuttal · Reviewer_GZc3 · 2026-04-01
> >
> > I thank the authors for fully addressing my concerns. I maintain my opinion that this paper deserves acceptance.

---

### Official Review · Reviewer_8698 · 2026-03-13

**Soundness:** 3
**Presentation:** 4
**Significance:** 3
**Originality:** 3
**Overall Recommendation:** 4
**Confidence:** 5

**Summary:**

The paper proposes a symmetry-based framework for curvature approximation in neural networks. The main idea is to use weight-space symmetries to build structured curvature approximations from a single gradient computation, with the chosen symmetry group controlling a trade-off between expressiveness and computational cost. A key contribution is the claim that particular restricted symmetry choices recover updates closely related to Shampoo and Muon. Empirically, the paper studies curvature quality on a small exact-Hessian setup and compares Symo (the obtained optimizer) with Muon and Adam on an MNIST autoencoder and a nanoGPT character-level benchmark.

**Compliance With Llm Reviewing Policy:**

Affirmed.

**Key Questions For Authors:**

- Can the authors provide direct runtime, FLOPs, memory, and optimizer-state comparisons against Muon and Shampoo? This would substantially strengthen the practical significance.

- Can the authors better justify the core approximation, either theoretically or with a targeted empirical study on approximation quality as orbit size changes?
- How sensitive are the Symo-Muon comparisons to the assumptions used in the experiments, especially where exact equivalence conditions are not fully enforced?

- Can the authors provide broader validation beyond the small exact-Hessian setting?

**Limitations:**

Yes

**Strengths And Weaknesses:**

## Strengths:
- S1: The framework is coherent and technically interesting. I found the symmetry-based view of curvature approximation genuinely novel and elegant (although I did not check in details relevant literature).
- S2: There is a nice conceptual contribution in showing how different symmetry choices induce different curvature structures and trade off cost versus approximation richness. This could be developed to find "optimal" structure under some assumptions in future work in my opinion.
- S3: The connection to Shampoo and Muon is elegant and gives the framework practical relevance beyond a purely theoretical construction.
- S4: The curvature-quality experiment is informative and gives some evidence that the approximation behaves reasonably, especially along optimization-relevant directions. However it would be even more convinving in higher dimensions (see W5).
- S5: The paper is well written, well structured, and easy to follow overall.

## Weaknesses
- W1: The core approximation step is not fully justified; the paper explicitly notes the lack of error bounds and leaves approximation accuracy as orbit size grows unclear.
- W2: The practical advantage over Muon/Shampoo is not convincingly demonstrated. The paper discusses tractability and computational savings, but does not report runtime, FLOPs, throughput, peak memory, or optimizer-state comparisons.
- W3: The experiments mainly support equivalence or competitiveness rather than a clear practical improvement over existing optimizers.
- W4: Some empirical comparisons do not satisfy all of the exact-equivalence conditions, so the practical validation is somewhat less clean than the theory might suggest.
- W5: The strongest analysis is still on relatively small-scale settings, which limits confidence about broader impact in larger practical regimes.

---

> ### Author Rebuttal · Authors · 2026-03-30
>
> We thank Reviewer 8698 for the thorough assessment. We are glad that the framework was found technically interesting, and that the connection to Shampoo/Muon was appreciated. Below, we address raised concerns.
>
> **W1 (Taylor expansion around the orbit average).** We agree that we have not provided theoretical bounds on the accuracy of our Taylor approximation, and how it varies with the symmetry group; to mitigate this, we conducted new experiments (see [Fig. 1](https://github.com/anonymous-reviewer-icml-2026/symo/blob/main/w-dist.pdf)) measuring approximation quality as a function of orbit size. Specifically, we measured the (normalized) distance between parameters and their orbit average, $\|w - w^\star\|^2 / \|w\|^2$. We used the same student-teacher four-layer MLP setup as in our Hessian similarity experiments (Section 4, Figure 4) and computed the relative distance for a fixed $w$ while varying the group size: [P;P;P], [P;I;P] and [I;P;I] (I being the identity group and P the permutation group). We ran 6 seeds and report the mean and 95% confidence interval.
> This confirms that, as the group grows, the distance between $w^\star$ and $w$ grows as well. We will include these results in the revised manuscript whilst noting that a full theoretical analysis remains to be carried out (but see below for a preliminary analysis). We will also emphasize that our framework's value is in providing a new route to flexible curvature approximation with controlled complexity, rather than a guarantee of point-wise Hessian accuracy.
>
> We derive a preliminary bound. Let $P \equiv \mathcal{R}_1(w, G)$ be a projection to the $G$-invariant subspace $V$. Then $w - w^\star = (I - P) w$. Since $P$ is an orthogonal projection, $\|w - w^\star\|^2 = \|w\|^2 - \|Pw\|^2$. By Lemma 3.1, for any group element $A$ (and using orthogonality): $\|Aw - w^\star\|^2 = \|A(w - w^\star)\|^2 = \|w - w^\star\|^2$, so every point on the orbit is equidistant from $w^\star$. The secant condition (Eq. 8b) is controlled by the Taylor remainder,
> $$\|\nabla L(Aw) - g^\star - H^\star(Aw - w^\star)\| \leq \frac{M}{2}\|w - w^\star\|^2,$$
> where $M$ is the Lipschitz constant of the Hessian. For $d \equiv \dim(w)$ and random initialization $w \sim \mathcal{N}(0, \sigma^2 I)$, the expected relative error becomes:
> $$\frac{\mathbb{E}\|w - w^\star\|^2}{\mathbb{E}\|w\|^2} = 1 - \frac{\dim(V)}{d}.$$
> Thus for smaller groups, $\dim(V)$ is larger and the Taylor approximation more accurate.
>
> **W2 (Experiments are too small and lack error estimates).** We will update plots to include means $\pm 95\\%$ confidence intervals over 5 random seeds for all training curves in Figures 4, 5 and 6 (see [Fig. 2](https://github.com/anonymous-reviewer-icml-2026/symo/blob/main/mnist-nanogpt.pdf), and [Fig. 3](https://github.com/anonymous-reviewer-icml-2026/symo/blob/main/similarity-std.pdf)). The variance across seeds is small, confirming that the observed equivalence is robust. We agree that MNIST and nanoGPT experiments are modest in scale. The primary contributions of this paper are conceptual/theoretical, with the introduction of a symmetry-based framework for estimating curvature and its connections to existing optimizers. Scaling to larger models is indeed an important direction for future work, and we will state this more explicitly in the conclusion.
>
> **Q (Runtime, FLOPs, memory comparisons?).** We agree that computational and memory comparisons with other optimizers would further strengthen the practical contribution. Symo's memory and computational costs depends highly on the chosen group size; however, the smallest (and therefore most "expensive") group that would make sense to use in practice is the one that yields the same complexity as Muon/Shampoo. In Figure 6 we showed that even with a much larger group (and therefore lower complexity), Symo goes a long way towards matching Shampoo's improvement over Adam. We therefore expect Symo to be a useful interpolant between Adam's low complexity and Shampoo's high performance. A detailed and meaningful comparison of runtime and memory costs would require more engineering time to ensure that e.g. matrix-vector products with each (often very low-complexity) term in the invariant representation of the Hessian are performed in parallel.
>
> **Q (Better justification of the core approximation?).** Please see W1 above.
>
> **Q (Sensitivity of Symo-Muon comparisons to assumptions?).** The conditions for exact equivalence are listed in Appendix I.1.2. Despite some practical differences (Items 3-6, using SVD instead of Newton-Schulz iteration, fan-in/fan-out scaling), we observe that training curves overlap closely.
>
> **Q (Validation beyond small settings?).** Please see our discussion of W2 above. Note that Figures 5 and 6 already show results on two different autoencoder benchmarks (MNIST and FACES) with multiple symmetry group configurations, and the nanoGPT experiment (10M~parameters) represents a step beyond toy settings.

---

> > ### Author Rebuttal · Reviewer_8698 · 2026-04-03
> >
> > Thanks for the rebuttal. I will keep my score of weak accept.

---

### Decision · Program_Chairs · 2026-04-30

**Decision:**

Accept (regular)

**Comment:**

*Motivation:* Exploiting the weight symmetry structure in neural networks to propose a better Hessian approximation.

*Contribution:* Proposing a novel Hessian approximation mechanism that partially explains the behavior of existing effective optimization methods such as Shampoo/Muon, while also offering an alternative to these methods.

*Reviews summary:* Reviewers found the contribution significant; in particular, they view the incorporation of weight equivalence structure into optimization as a fundamental advance. However, the current derivations frequently rely on neglecting Taylor expansion errors, which may be substantial in the analysis. Additionally, the notion of averaging and its connection to gradient descent convergence is insufficiently explained. One reviewer remains unconvinced about the claimed connection to Shampoo.
Rebuttal summary: While the rebuttal adequately addresses most reviewer concerns, a few remain unresolved, including the connection to Shampoo and the magnitude of the Taylor expansion error.

*AC review:* While the paper makes an interesting contribution, the AC believes it would benefit from a minor revision.
Although the reviews are positive, the paper has two issues requiring further clarification:

- (1) The paper's original goal is to approximate the Hessian of a non-convex function; yet the proposed approximation is constrained to be PSD. Approximating a non-PSD matrix with a PSD one requires formal justification for two reasons: (i) no approximation metric is defined, leaving it unclear how such an approximation should be measured; and (ii) explaining why methods such as Shampoo/Muon leverage curvature is non-trivial, as the authors must demonstrate how these methods approximate the Hessian with a PSD matrix — for instance, it is not established whether Shampoo's preconditioning matrix shares eigenvectors with the Hessian. It is worth noting that while using a PSD preconditioner is well-motivated, as acknowledged in the paper and in [Bottou et al., 2018], this does not imply that the Hessian or the underlying curvature is PSD, nor that these methods employ an approximate Hessian.

- (2) As raised by reviewers 8698 and gmTt, the Taylor expansion error in the analysis may be excessively large, since  \( \| w^* - w\| \) can be arbitrarily large under the authors' definition. Consequently, the analysis contains an uncontrolled error term that is neither theoretically bounded nor analytically characterized, and this error may propagate through all subsequent steps.

*Summary of AC-reviewer discussions.* I discussed the issue (1) with the reviewers. We discussed whether the paper's notion of curvature may differ from the Hessian — for instance, it could reflect a specific preconditioning mechanism approximating the Gauss–Newton preconditioner. In particular, one reviewer suggested that the authors may be dropping certain terms from the Hessian in akin to the Gauss–Newton method. Nevertheless, the current manuscript states in multiple places, including Figure 1, that the explicit goal is Hessian approximation.  Furthermore, Taylor approximation in the derivations are based on Hessian not an alternative curvature notion. Another reviewer argued that Hessian approximation is up to sign of eigenvalues which is not proven or even discussed in the current version of the paper.  During the discussion phase, a high-scoring reviewer acknowledged that optimization fell outside their area of expertise, prompting me, as AC, to review the paper directly before reaching a decision — as there was insufficient time to assign an emergency reviewer following the AC-reviewer discussion. I did my best to provide constructive feedback to guide the authors in strengthening their results given the limited time available that I had.

*Decision:* Based on discussions with SAC and reviewers, AC recommend weak acceptance.